# Better Practices for Domain Adaptation

**Linus Ericsson**[1]  **Da Li**[2]  **Timothy M. Hospedales**[1,2]

[1]University of Edinburgh
[2]Samsung AI Center Cambridge

**Abstract**  Distribution shifts are all too common in real-world applications of machine learning. Domain adaptation (DA) aims to address this by providing various frameworks for adapting models to the deployment data without using labels. However, the domain shift scenario raises a second more subtle challenge: the difficulty of performing hyperparameter optimisation (HPO) for these adaptation algorithms without access to a labelled validation set. The unclear validation protocol for DA has led to bad practices in the literature, such as performing HPO using the target test labels when, in real-world scenarios, they are not available. This has resulted in over-optimism about DA research progress compared to reality. In this paper, we analyse the state of DA when using good evaluation practice, by benchmarking a suite of candidate validation criteria and using them to assess popular adaptation algorithms. We show that there are challenges across all three branches of domain adaptation methodology including Unsupervised Domain Adaptation (UDA), Source-Free Domain Adaptation (SFDA), and Test Time Adaptation (TTA). While the results show that realistically achievable performance is often worse than expected, they also show that using proper validation splits is beneficial, as well as showing that some previously unexplored validation metrics provide the best options to date. Altogether, our improved practices covering data, training, validation and hyperparameter optimisation form a new rigorous pipeline to improve benchmarking, and hence research progress, within this important field going forward.

## 1 Introduction

Supervised deep learning models achieve impressive results when training and testing data are identically distributed. However, perhaps the main failure mode of computer vision and pattern recognition systems in practice is due to the near-ubiquitous distribution shift between data curated for model training, and real-world data encountered during deployment [6]. This distribution shift issue has motivated a tremendous amount of work in the area of unsupervised domain adaptation (UDA) [6]. UDA methods aim to alleviate domain shift by collecting freely available unlabelled data during deployment to a target domain and adapting vision models based on this unlabelled data.

Hundreds of unsupervised adaptation algorithms have now been proposed based on various principles from distribution alignment [23], to domain adversarial learning [10] and much more. However, without exception, a key challenge for every one of these algorithms is: *how do we tune hyperparameters and conduct model selection?* In conventional supervised learning, hyperparameters and model selection (stopping criteria) are handled systematically by maximising accuracy on a validation split of the training set. In unsupervised domain adaptation there is no such straightforward solution because the target domain has no labels with which to compute accuracy, and the source domain is not representative of the target domain.

Despite the importance of this issue—upon which any practical application of domain adaptation hinges—there has been relatively little systematic study of validation protocols and algorithms for UDA [48, 7, 35]. Worse, a recent meta-review and re-evaluation of the domain adaptation literature found that most published code did not use consistent or fair model selection criteria [26], and furthermore when evaluated under consistent and fair model selection criteria most existing results

can not be replicated [26]. This mini "replication crisis" in domain adaptation highlights the need for studying validation protocols for UDA, and for fair benchmarking to drive reliable progress.

The few existing fair model selection criteria for UDA are based on diverse intuitions such as simply applying UDA algorithm objectives on the validation split of the unlabelled target set, priors on the expected distribution of labels [7, 35], or relying on the validation accuracy in the source domain [48]. However there is little first principles justification to pick among these reasonable intuitions, and there is little empirical evaluation to understand which are best, and how close they come to the performance of an oracle validator, which has been the basis of many reported results in the literature [26].

These challenges exist throughout the domain adaptation literature. They arise across all three popular branches of adaptation for recognition: Unsupervised Domain Adaptation (UDA) [10, 39, 42], Source-Free Domain Adaptation (SFDA) [20, 47] and Test Time Adaptation (TTA) [45, 22]. They also arise across different kinds of domain adaptive learning problems from classification [10] to regression [5], dense prediction [49], and detection [16].

The lack of a clear validation criterion for DA is an obstacle to its practical application. As an example, AutoML is a field with great potential to automate machine learning tasks for real-world applications [15]. But in order to automate anything (e.g. algorithm, hyperparameter, checkpoint selection), we need a metric to optimise. In the case of supervised learning, this metric is naturally validation performance on an unseen labelled set. However, the choice of metric is not straightforward for UDA/SFDA/TTA due to the lack of labels. A major contribution of this paper is to clarify what such an optimisation metric should be for domain adaptation, thereby laying the foundations that allow bringing AutoML to DA.

To address this issue, we conduct a large-scale benchmark of 10 domain adaptation algorithms with 15 different validation criteria and three DA settings (UDA, SFDA, TTA). We identify which DA validators can be applied to each setting, characterise the size of the challenge in each case in terms of the gap between practically achievable and best-case DA performance, and identify the best existing validators. We identify effective practices in terms of using validation splits to estimate target performance. We highlight the risk of adaptation failure in SFDA and TTA as a likely fatal blocker for deployment in practice as existing validators do not reliably prevent this. These results should drive future practice both in DA research – which should use these validators, rather than unrealistic oracle HPO; and in validator research – which should aim to develop validators which surpass the best that we report.

## 2 Related Work

### 2.1 Domain Adaptation

There are now too many domain adaptation algorithms to review here, and we refer the reader to good surveys such as [6, 28]. Most deep UDA algorithms proceed by performing supervised learning on the source domain data, and some kind of unsupervised objective on the target domain data. Representative families of approach include objectives that penalise misalignment between the source and target domain feature distributions [23], train a domain classifier that can then be used adversarially to penalise distinguishable source and target domain features [10], or penalise deviation from a prior on the expected target label distribution [37]. However, all algorithms have a number of hyperparameters, such as stopping iteration and strength of the weighting factor for supervised vs unsupervised loss components. How to set these hyperparameters is not clear given the lack of a labelled target domain validation set in UDA applications.

The long-established mainstream setting for unsupervised domain adaptation (UDA) assumes that source and target data are accessed simultaneously for training. Two related problem variants have more recently gained rapid popularity, namely source-free domain adaptation (SFDA) and Test Time Adaptation (TTA). SFDA refers to the condition where pre-trained source models should

be adapted to the target data without revisiting the source data [20] – for example, by unsupervised fine-tuning. TTA [45, 40] similarly adapts a pre-trained model without access to the source data, but assumes that the test data arrives in mini-batches, providing the opportunity to adapt to each mini-batch before making decisions on their labels. The newer SFDA and TTA have both rapidly gained traction as being more "practical" in an era of pre-trained models [3]. However, algorithms for both of these settings still have many hyperparameters (e.g., learning rate, number of iterations, regularisation strengths), and hence suffer from the lack of a clear validation protocol in a DA context. Most of the seminal studies in this area do not show valid HPO criteria in their papers or code.

## 2.2 Validation Approaches for DA

Comparatively few papers have systematically studied validation criteria for UDA, given the importance of this issue for its practical application. Typical solutions applied by UDA algorithm papers include: (1) Oracle risk. Many papers use the target test set for hyperparameter selection [26], which is obviously incorrect as it can not be used in real applications; (2) Source risk. Evaluating the adapted model on the source validation set is reasonable but may not be a good validation criterion due to domain shift between source and target domains; (3) Evaluating another UDA algorithm objective (such as InfoMax [37] and MMD [23]) on an unlabelled validation split of the target set; (4) Validation domain. Use of a held-out labelled validation domain, as used in the VisDA challenge [30], is fair. However, this assumes multiple labelled domains, which may not be available in practice, and also raises additional questions of whether the optimal hyperparameters for the validation domain are representative of the optimal hyperparameters for the target domain.

Besides the above strategies, a few purpose-designed validation criteria have been proposed: Deep embedded validation (DEV) [48] weights the source validation risk by the probability that each sample belongs to the target domain. Meanwhile, Silhouette score [32], batch nuclear-norm minimisation (BNM) [7], and soft neighbourhood density (SND) [35] criteria boil down to evaluating the adapted models' posterior label distribution on the target domain under different notions of a prior for the expected target domain label distribution. Mean ensemble-based validation (ENS) [32] considers a linear combination of the above criteria. However, overall it is unclear which to prefer for DA.

## 2.3 Benchmarking Domain Adaptation

There have been two major benchmarking exercises in UDA. The VisDA competition challenge [30] provides a labelled validation domain for model selection and hyperparameter optimisation (HPO). However, validation domains may not be available in practice – and if they are, they may not be representative of the target domain. Thus, the vast majority of research literature on UDA has not used this approach. A recent empirical evaluation [26, 25] analysed the GitHub repositories of a number of UDA methods and found that: (1) In practice different methods used very different validation criteria for empirical evaluation, making published results incomparable with each other; (2) A large number of prior studies used the oracle risk as a validation criterion, meaning that their results are not representative of how well domain adaptation would work in reality using validation criteria that can be implemented in practice; (3) Variation in existing validation criteria was high compared to variation across adaptation algorithms, and none of them was strongly correlated with recognition performance. Our evaluation extends this early study but goes beyond it in considering all three major branches of DA research (UDA, SFDA, TTA), exploring a wider variety of validators, and demonstrating how validator performance can be improved through proper use of validation splits within the target domain.

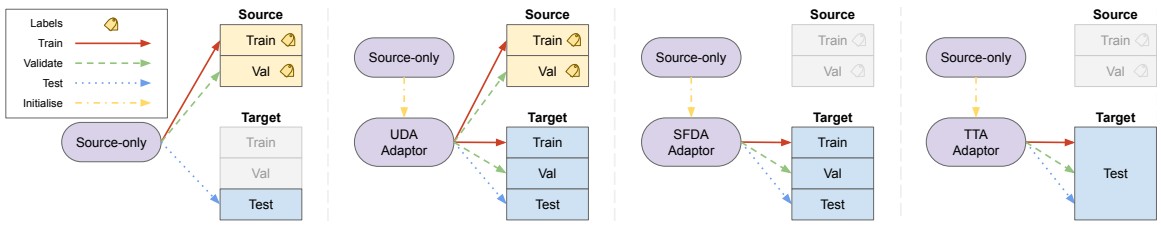

Figure 1: How the source and target domains are split and how each split is used for (1) the source-only model (2) UDA adaptors, (3) SFDA adaptors and (4) TTA adaptors.

## 3 Background

### 3.1 Problem Setup

**Unsupervised Domain Adaptation**: In the UDA setup, one typically trains a model $f_{\boldsymbol{\theta}} : \mathcal{X} \mapsto \mathcal{Y}$ on a labelled dataset, $\mathcal{D}_S = \{\boldsymbol{x}_i, \boldsymbol{y}_i\}_{i=1}^{N_S}$, consisting of data sampled from a source domain, $p_S$. The goal is then to adapt $f_{\boldsymbol{\theta}}$ using an unlabelled dataset, $\mathcal{D}_T = \{\boldsymbol{x}_i\}_{i=1}^{N_T}$, sampled from a target domain, $p_T$. The general learning objective to be minimised w.r.t. to $\boldsymbol{\theta}$ can be simplified as follows,

$$L(f_{\boldsymbol{\theta}}, \mathcal{D}_S, \mathcal{D}_T) = L_{\mathrm{sup}}(f_{\boldsymbol{\theta}}, \mathcal{D}_S) + L_{\mathrm{da}}(f_{\boldsymbol{\theta}}, \mathcal{D}_S, \mathcal{D}_T), \tag{1}$$

where $L_{\mathrm{sup}}(\cdot)$ could be cross-entropy loss for classification and mean square error for regression problems, and $L_{\mathrm{da}}(\cdot)$ is the adaptation loss, such as MMD [42], CORAL [39] and DANN [10] losses.
**Source-Free Domain Adaptation**: The SFDA setting aims to adapt a pre-trained source domain model to the target domain, relaxing the assumption of joint occurrence of source and target domain data in UDA. So first, a source model will be optimized using source domain data: $\hat{\theta} = \arg\min_{\theta} L_{\mathrm{sup}}(f_{\boldsymbol{\theta}}, \mathcal{D}_S)$. Then the trained source model $\hat{\theta}$ will be adapted to the target domain by

$$\theta^* = \arg\min_{\hat{\theta}} L_{\mathrm{sfda}}(f_{\hat{\theta}}, \mathcal{D}_T). \tag{2}$$

where now $L_{\mathrm{sfda}}$ is an unsupervised loss, such as clustering [47] or information maximization [20].
**Test-Time Adaptation**: Unlike SFDA, TTA assumes the batch-wise target domain data $X \sim \mathcal{D}_T$ comes in a stream and adapts a pre-trained source model for each minibatch $X$ as

$$\theta^* = \arg\min_{\hat{\theta}} L_{\mathrm{tta}}_{X \sim \mathcal{D}_T} (f_{\hat{\theta}}, X), \tag{3}$$

where $L_{\mathrm{tta}}$ is commonly the unsupervised loss, such as self-supervised learning and entropy minimisation losses, which could be essentially similar to $L_{\mathrm{sfda}}$.

### 3.2 Model Selection

Due to the lack of target domain labels in the various domain adaptation settings we consider, the model selection process must proceed as follows. Given a set of candidate models, as configured by hyperparameters $\boldsymbol{h} \in \mathbb{H}$, where $\mathbb{H}$ is the pool of hyperparameter sets, the best candidate model is selected based on its evaluation score, $d(f_{\boldsymbol{\theta}}, \mathcal{D}_V)$[1], where $\mathcal{D}_V$ is a validation dataset. The process can be formalised as

$$\boldsymbol{h}^* = \arg\max_{\boldsymbol{h}} \ d(f_{\boldsymbol{\theta}_{\boldsymbol{h}}^*}, \mathcal{D}_V),$$
$$\text{s.t.} \quad \boldsymbol{\theta}_{\boldsymbol{h}}^* = \arg\min_{\boldsymbol{\theta}} \ L(f_{\boldsymbol{\theta}}, \mathcal{D}_S, \mathcal{D}_T; \boldsymbol{h}). \tag{4}$$

---

[1]Assuming the model performance is a monotonically decreasing function of the output of $d(\cdot, \mathcal{D}_V)$.

Table 1: A summary of the adaptation algorithms considered

| | Algorithm | Approach |
|---|---|---|
| UDA | ATDOC [21] | Pseudo-labelling |
| | BNM [7] | SVD loss |
| | DANN [10] | Adversarial |
| | MCC [17] | Information maximisation |
| | MCD [36] | Classifier discrepancy |
| | MMD [23] | Feature distance |
| SFDA | AAD [47] | Clustering |
| | NRC [46] | Graph clustering |
| | SHOT [20] | Information maximisation |
| TTA | SHOT [20] | Information maximisation |
| | TENT [45] | Entropy minimisation |

Table 2: A summary of the validators considered.

| Criterion | Approach |
|---|---|
| RankMe [11] | Rank estimation |
| AMI [25, 32] | Cluster quality |
| ARI [31] | Cluster quality |
| V-Measure [33] | Cluster quality |
| FMI [9] | Cluster quality |
| Silhouette [25, 32] | Cluster quality |
| DBI [8] | Cluster quality |
| CHI [4] | Cluster quality |
| BNM [7] | Label prior |
| MMD [23] | Domain Alignment |
| CORAL [39] | Domain Alignment |
| SND [35] | Label prior |
| InfoMax [37] | Label prior |
| Entropy | Label prior |
| Source Accuracy | Source accuracy |

However, two things in UDA complicate this process: 1) choice of the validation set $\mathcal{D}_V$; and 2) definition of the evaluation metric $d(\cdot, \cdot)$ when $\mathcal{D}_V = \{x_i\}_{i=1}^{N_V}$ is an unlabelled set.

Several validators have been proposed in the literature, such as SND [35], BNM [7] and DEV [48]. Additionally, it is worth remarking that popular DA losses such as IM [20], and Entropy [44, 24], can also be used as validators. We explore a large number of validators in addition to these, including those based on domain alignment, like MMD [42] and CORAL [39], clustering [33] and feature matrix rank [11]. The full list of validators we consider in shown Tab. 2 with full details in Appendix D.

## 4 Evaluation

Our evaluation extends the benchmark of [26]. We make their setup more rigorous by splitting the target domain into train/val/test sets. Previous works often compute target performance on the same data that the algorithms adapt to or the same data that the validators use. This fails to properly measure generalisation performance as we will show later. Our splits and how we use them are detailed in Figure 1.

In order to compare different validation criteria, we train a large number of models across several datasets, algorithms and hyperparameter choices. We want the optimal validator to behave similarly to the target domain test performance of the corresponding algorithm. We measure the quality of each validator in two ways: 1) computing the Spearman rank correlation between validator scores and oracle test accuracy, 2) using the validator to select the best model for an algorithm/task pair and comparing the test performance of it against the best model as selected by the oracle.

**Questions**: Through the experimental evaluation below on three different settings, we aim to answer the following questions: (i) *Are the validation criteria sufficiently good to drive HPO and model selection in UDA?* We also extend this question to regression problems in Appendix A. (ii) *What is the impact of validating on the training set versus an independent validation split?* (iii) *Are the observations still consistent when source data is absent during adaptation (SFDA), and when we must adapt to the test-set itself (TTA)?*

### 4.1 Unsupervised Domain Adaptation

#### 4.1.1 Setup. In this section we describe our evaluation procedure for UDA.

**Datasets**: We use a wide range of UDA benchmark datasets: MNIST-M [10] which consists of a domain shift from standard MNIST [19] to a modified version; The VisDA-2017 [29] dataset which

Table 3: Comparison of validation criteria for model selection in UDA. Averages over all 21 domain transfers evaluated. We report (i) the target test performance for the top models selected by each validator, and (ii) the correlation coefficient between the validator scores and the test performance over all hyperparameters and checkpoints. The colour of a cell indicates whether that model/validator combination beats the source-only model (green) or not (red).

| | RankMe | AMI | ARI | V-Measure | FMI | Silhouette | DBI | CHI | BNM | MMD | CORAL | SND | IM | Entropy | Accuracy | Oracle |
|---|---|---|---|---|---|---|---|---|---|---|---|---|---|---|---|---|
| ATDOC | 58.24 | 67.70 | 67.71 | 67.79 | 67.71 | 46.73 | 49.55 | 16.55 | 64.29 | 52.46 | 55.96 | 24.13 | 64.61 | 61.23 | 68.06 | 72.24 |
| BNM | 61.36 | 69.32 | 69.48 | 69.29 | 69.48 | 62.42 | 51.58 | 33.10 | 66.88 | 52.64 | 60.92 | 47.70 | 67.01 | 65.98 | 66.02 | 71.09 |
| DANN | 62.00 | 64.76 | 64.35 | 64.55 | 63.23 | 56.06 | 53.86 | 36.89 | 62.72 | 51.62 | 60.61 | 46.51 | 62.79 | 62.83 | 62.44 | 68.27 |
| MCC | 62.36 | 69.65 | 70.06 | 69.66 | 69.68 | 63.21 | 40.48 | 24.72 | 66.84 | 55.12 | 54.66 | 35.13 | 66.79 | 65.28 | 69.11 | 72.41 |
| MCD | 60.80 | 60.31 | 46.45 | 60.26 | 31.23 | 16.54 | 28.58 | 8.99 | 63.83 | 51.06 | 47.44 | 13.66 | 64.43 | 56.44 | 63.83 | 67.75 |
| MMD | 60.37 | 65.98 | 63.56 | 66.00 | 63.56 | 54.83 | 51.93 | 35.22 | 61.37 | 46.66 | 58.41 | 40.08 | 61.57 | 61.06 | 63.83 | 67.44 |
| Avg. | 60.86 | 66.29 | 63.60 | 66.26 | 60.81 | 49.96 | 46.00 | 25.91 | 64.32 | 51.59 | 56.33 | 34.54 | 64.53 | 62.14 | 65.55 | 69.87 |
| Avg. Rank | 8.50 | 3.33 | 3.92 | 3.00 | 4.42 | 11.00 | 12.33 | 15.00 | 6.00 | 11.33 | 10.33 | 14.00 | 5.17 | 7.33 | 4.33 | - |
| Correlation | 0.29 | 0.62 | 0.62 | 0.65 | 0.58 | 0.01 | -0.40 | -0.60 | 0.35 | 0.30 | -0.47 | -0.15 | 0.36 | 0.30 | 0.50 | - |
| Source-only | 63.58 | 49.81 | 48.04 | 48.02 | 48.04 | 47.88 | 32.62 | 46.35 | 54.90 | 63.69 | 49.31 | 32.49 | 49.30 | 49.22 | 63.48 | 65.60 |

contains *train*, *validation* and *test* domains — we consider the shifts `train → validation` and `train → test`; Office-31 [34] which consists of three domains: *amazon*, *dslr* and *webcam*; and Office-Home [43] with four domains: *art*, *clipart*, *product* and *real*. In total, we consider 21 different domain shifts.

**Adaptation Algorithms & Validators**: We consider six representative domain adaptation algorithms, spanning both recent and classic methods and a variety of underlying principles. These include the pseudo-label based **ATDOC** [21]; domain-adversarial learning with the seminal **DANN** [10]; domain-alignment with **MMD** [23]; **BNM** and **MCC** which optimise the target label distribution under nuclear norm prior and minimum class confusion priors respectively, and the classifier-discrepancy-based **MCD** [36]. We explore tuning these models with a large number of potential validation criteria as listed in Table 2.

We start by finetuning a network on the source task. We take ResNet50 weights pretrained on ImageNet [14] for all datasets apart from MNIST-M where a smaller CNN is used. The final classification layer is replaced by an MLP head consisting of two blocks of {Linear, ReLU, Dropout} followed by a final linear layer. We finetune only this head on the source task using a standard categorical cross-entropy loss. 10 models are trained with learning rates sampled uniformly at random from a logarithmic scale between $10^{-5} - 10^{-1}$. These runs form the set of checkpoints for the *source-only* model. They are not used for evaluating the validation criteria, but we report performances at times for comparison. When training each adaptation algorithm, we use the source-only model weights as initialisation for both the backbone and MLP head. The specific source-only checkpoint used as initialisation is the one with the highest source validation accuracy and in case of ties we select the checkpoint trained for the fewest amount of epochs.

For each of our 6 adaptation algorithms, we sample 10 sets of hyperparameters and train one model per set. The training uses both source and target data. The number of epochs depends on the dataset and specific target domain, but in all cases, we save 20 checkpoints during the course of training. The optimizer is Adam with parameters {`betas=(0.9, 0.999)`} and the weight decay is always 0.0001. The learning rate is always part of the sampled hyperparameters, and it is updated during training via cosine annealing with a warmup phase during the first 5% of training. The full details of our training procedure and hyperparameter search spaces can be found in Appendix C.

**4.1.2 Results**. The results in Table 3 report the performance of each adaptation algorithm and validation criterion combination, averaged over all 21 domain transfer tasks – in terms of both test accuracy after HPO and the weighted Spearman correlation coefficient between validation scores and testing accuracy. More detailed correlation plots are given in Appendix E. Table 4 shows how easily tunable the algorithms are, via the percentage of all algorithm checkpoints outperforming the baseline.

Table 4: Percentage of all algorithm checkpoints which outperform the baseline source-only model, in the UDA setting. The ATDOC, BNM and MCC algorithms are the most easily tunable, with over 40% of hyperparameter choices leading to better-performing models. We also see that when selecting checkpoints with the oracle validator, 18.3% of the source-only checkpoints outperform the one selected as the baseline using the source validation accuracy validator.

| Source-only | ATDOC | BNM | DANN | MCC | MCD | MMD |
|---|---|---|---|---|---|---|
| 18.3 | 42.8 | 48.4 | 24.9 | 43.4 | 14.6 | 29.5 |

**How well does unsupervised validation work?** From Table 3 we can draw a rich set of observations: (1) The validation criteria have varying ability to predict the test accuracy and thus drive HPO in domain adaptation. This is visible in the correlation scores, ranging from -.60 to .65 correlation coefficient at best; and the significant variability of testing performance when using the various criteria to drive HPO. Importantly, it is also visible in the gap between the performance of the best criteria and the best-case oracle criterion. (2) The best validation criterion is the previously un-studied V-measure score, which has the best average rank of 3.0 across all the validators, and closes 70% of the gap between the baseline of 63.5% and oracle upper bound of 72.4% when paired with the MCC adapter. (3) The DA algorithms themselves vary substantially in how easy they are to tune, with MCD and ATDOC for example being highly dependent on choice of validator, versus BNM which is comparatively insensitive to the choice of validator. Practitioners may prefer to opt for comparatively easy to tune adaptation algorithms, given the challenge of validation for DA.

**What is the impact of validating on training vs validation splits?** An important design choice in validation is which data split the validator is evaluated on. As discussed in [26, 25], while prior work that validates on the source domain has fairly consistently used the source validation set; prior work that validates on the unlabelled target domain has been inconsistent in the choice between validating using the train or an independent val split. Since learning is driven by applying an adaptation loss on the target train set, there is the possibility of overfitting during unsupervised adaptation. Thus we conjecture that one should validate on a disjoint split of the target domain. In Tab. 3, we avoided this issue by taking the best split for each validator. We now analyse this issue by comparing using the train vs validation split for evaluating criteria. From the results in Fig. 2 we see that for all the top-performing criteria the val split is preferred. While this result might seem unsurprising in retrospect, we emphasise that the use of a val split is NOT standard practice in the literature, even in thorough recent evaluations [25]. We show that there is thus a trade-off between using held-out validation data to improve the validators, and using as much training data as possible to improve adaptation.

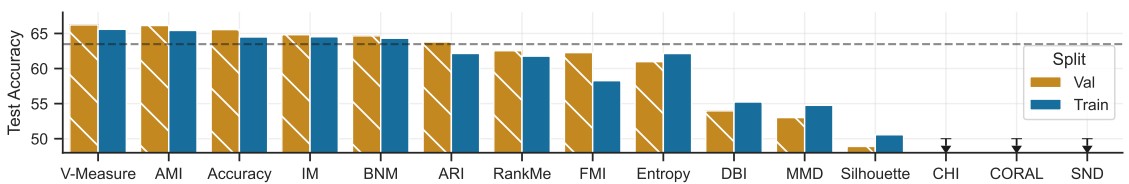

Figure 2: Comparison of split for evaluation of validation criteria. We report the average target test accuracy of selected models for each validator when applied on (blue) target train data and (orange) target validation data. The dashed line is the source-only model performance.

Table 5: Comparison of validation criteria for model selection in SFDA on Office-Home. We report (i) the target test performance for the top models selected by each validator, and (ii) the correlation coefficient between the validator scores and the test performance over all hyperparameters and checkpoints. For each algorithm, we include the source-only checkpoint in the pool available to validators (indicated by the "+SO" suffix). The colour of a cell indicates whether that model/validator combination beats the source-only model (green) or not (red), with a darker red colour meaning it fails to achieve half of the source-only model performance.

| | RankMe | AMI | ARI | V-Measure | FMI | Silhouette | DBI | CHI | BNM | SND | IM | Entropy | Accuracy | Oracle |
|---|---|---|---|---|---|---|---|---|---|---|---|---|---|---|
| AAD+SO | 61.63 | 57.41 | 1.60 | 60.19 | 1.59 | 1.74 | 53.62 | 1.90 | 62.51 | 56.29 | 59.40 | 4.78 | - | 65.71 |
| NRC+SO | 57.30 | 58.44 | 6.74 | 62.73 | 6.70 | 1.62 | 45.12 | 1.70 | 58.18 | 56.84 | 57.71 | 40.70 | - | 64.93 |
| SHOT+SO | 59.20 | 59.73 | 60.88 | 60.99 | 59.38 | 57.54 | 55.41 | 46.72 | 54.14 | 57.13 | 54.52 | 59.51 | - | 64.04 |
| Avg. | 59.38 | 58.52 | 23.07 | 61.30 | 22.56 | 20.30 | 51.38 | 16.77 | 58.28 | 56.75 | 57.21 | 35.00 | - | 64.89 |
| Avg. Rank | 4.33 | 3.33 | 7.33 | 1.67 | 9.00 | 9.67 | 7.67 | 10.67 | 5.00 | 6.67 | 6.00 | 6.67 | - | - |
| Correlation | -0.02 | 0.11 | -0.32 | 0.09 | -0.08 | -0.54 | 0.01 | -0.77 | -0.01 | 0.06 | 0.02 | -0.11 | - | - |
| Source-only | - | - | - | - | - | - | - | - | - | - | - | - | 56.49 | - |

## 4.2 Source-free Domain Adaptation

**4.2.1 Setup**. For SFDA we use the Office-Home dataset as a benchmark, covering all 12 domain shifts. The same source-only models that we produced for UDA are also used here for initialisation of the same architecture. Three recent SFDA algorithms adapt the model on target domain data, AAD [47], NRC [46] and SHOT [20]. For each algorithm, we sample 10 sets of hyperparameters and train for 200 epochs. The setup follows the UDA setting described above, with the main difference being the adaptation algorithms and validators only have access to target data (Fig. 1). As the source domain is not available in this setting, we can only apply our validators to the target domain splits. This means CORAL and MMD are not applicable, since they need both domains to compute their scores. Following our results in Section 4.1.2 we use the target validation split for all validators as the source data is absent in this case.

**4.2.2 Results**. Analogous to UDA, the results in Table 5 report the performance of each adaptation algorithm and validation criterion combination, averaged over all 12 domain transfer SFDA tasks – in terms of both test accuracy after HPO and the weighted Spearman correlation coefficient between validation scores and testing accuracy. More detailed correlation plots are given in Appendix E.

**How does unsupervised validation work in the absence of source data?** From the results in Table 5 we can draw a set of conclusions analogously to UDA. Specifically, (i) Here, the best validators are RankMe and V-measure, with V-measure closing up to 75% of the gap between the baseline and oracle when combined with NRC. (ii) However the AAD and NRC algorithms are highly sensitive to validator choice, with the weaker validators such as FMI and CHI producing catastrophically poor performance, suggesting that SHOT might be preferred in practice even though NRC has the best accuracy when paired with its preferred validator, and AAD when validated with the oracle. (iii) Many algorithm-validator combinations lead to *worse* performance than the baseline source-only model. This highlights an important point that in the absence of highly reliable validation criteria, DA algorithms pose a risk of making the performance even worse. This issue is one which is not widely analysed in academic DA but is obviously crucial. Please note that we also included the model initialization (i.e. the source-only model) as one of the checkpoints available for selection by the criteria. However, many validators fail to detect adaptation failure and select a safe source-only model.

## 4.3 Test-Time Adaptation

**4.3.1 Setup**. We next adopt the TTA setting, where a pre-trained model adapts to the test data as it comes, one batch at a time. For simplicity, we use the episodic setting [45] where the model is reset after each batch. We use the most common TTA benchmark of CIFAR10-C, consisting of 15 versions of

Table 6: Test-Time Adaptation on CIFAR10-C, at corruption level 5. We use the episodic setup where the model is reset after each batch. For each algorithm, we include the source-only checkpoint in the pool available to validators (indicated by the "+SO" suffix). The colour of a cell indicates whether that model/validator combination beats the source-only model (green) or not (red).

| | RankMe | AMI | ARI | V-Measure | FMI | Silhouette | DBI | CHI | BNM | SND | IM | Entropy | Accuracy | Oracle |
|---|---|---|---|---|---|---|---|---|---|---|---|---|---|---|
| SHOT+SO | 78.38 | 37.37 | 36.85 | 37.69 | 36.86 | 36.50 | 38.60 | 43.87 | 45.22 | 53.48 | 46.80 | 39.14 | - | 86.24 |
| TENT+SO | 84.06 | 84.14 | 84.17 | 84.13 | 84.17 | 84.23 | 81.84 | 75.98 | 84.22 | 79.89 | 84.21 | 83.20 | - | 84.76 |
| Avg. | 81.22 | 60.76 | 60.51 | 60.91 | 60.51 | 60.36 | 60.22 | 59.93 | 64.72 | 66.68 | 65.51 | 61.17 | - | 85.50 |
| Avg. Rank | 4.50 | 7.50 | 7.75 | 7.50 | 7.25 | 6.50 | 8.50 | 8.50 | 3.00 | 6.50 | 3.00 | 7.50 | - | - |
| Correlation | 0.10 | 0.13 | 0.14 | 0.13 | 0.14 | 0.13 | -0.56 | 0.09 | 0.09 | -0.55 | 0.08 | -0.02 | - | - |
| Source-only | - | - | - | - | - | - | - | - | - | - | - | - | 70.64 | - |

the CIFAR10 test set with various corruptions applied, including Gaussian noise, pixelation and fog. Additionally, we investigate whether existing TTA algorithms are able to deal with the more complex distribution shifts from Office-Home, using all 12 domain shift setups. For CIFAR10-C, we use the pre-trained CIFAR10 checkpoint of [22] as our source-only model and initialisation for the TTA algorithms. For Office-Home, we use the same source-only checkpoints as in the UDA and SFDA sections above. Two algorithms are trained: SHOT [20] which uses information maximisation and pseudo-labelling to align target representations and TENT [45] which adapts by minimising the entropy of its predictions on the test batch. As this setting only exposes a single batch to the model at a time, both training and validation use the same data. As in the SFDA setting, CORAL and MMD are not applicable, since they need both domains to compute their scores. This also means that there is no validator based on accuracy, as it requires source data to be computed.

**4.3.2** **Results**. Tables 6 and 7 report the performance of each adaptation algorithm and validation criterion combination averaged over all CIFAR-C and Office-Home TTA tasks.

**Is Test-Time Adaptation effective when performing proper model selection?** The results for CIFAR in Table 6 lead to a different conclusion from that of UDA and SFDA. (i) RankMe is again the best validation criterion, and interestingly, we see that the top validators now manage to almost match the oracle performance. (ii) TENT is robust to the choice of validator, with consistently good performance close to oracle. SHOT obtains reasonable performance only when RankMe validator is used.

While TENT-based TTA plus various validators above show a success case for good practice adaptation on CIFAR10-C, we next ask whether these good results persist to a real rather than synthetic adaptation task. Table 7, shows the results of the Office-Home benchmark. From the results, we can see that: (1) There is only a 2-3% gap between the oracle best case and the baseline, suggesting that all algorithms struggle on this benchmark, even for best-case HPO. (2) Almost all algorithm-validator combinations are worse than the 57% source-only accuracy, similar to the SFDA case discussed earlier. When we compare the adaptation performance with- and without- access to the source-only model in the pool of checkpoints for HPO, SHOT has little improvement. The validators are not able to respond to the destructive adaptation and fail to pick a safe pre-adaptation model. Thus, we suggest that the strong success of TTA methods on synthetic benchmarks may not be representative of real-world adaptation problems, especially when required to use fair validation.

## 5  Conclusion

In this work, we performed a comprehensive study of HPO and model selection for domain adaptation, covering 10 algorithms and 15 validators across three settings (UDA, SFDA and TTA). We have found that previously unexplored validators like RankMe and V-Measure perform well across several settings, but the optimal validator is setting and algorithm dependent. Thus practitioners may wish to consider tuning sensitivity as a key factor for algorithm selection beyond reported

Table 7: Test-Time Adaptation on Office-Home. We use the episodic setup where the model is reset after each batch. Algorithms that include the source-only checkpoint in the pool available to validators are marked by the suffix "+SO". The colour of a cell indicates whether that model/validator combination beats the source-only model (green) or not (red), with a darker red colour meaning it fails to achieve half of the source-only model performance.

| | RankMe | AMI | ARI | V-Measure | FMI | Silhouette | DBI | CHI | BNM | SND | IM | Entropy | Accuracy | Oracle |
|---|---|---|---|---|---|---|---|---|---|---|---|---|---|---|
| SHOT | 20.46 | 9.38 | 10.09 | 9.43 | 10.09 | 14.07 | 34.67 | 12.06 | 9.10 | 35.58 | 10.13 | 7.52 | - | 59.05 |
| TENT | 42.40 | 37.57 | 42.79 | 43.49 | 40.77 | 2.25 | 39.90 | 3.61 | 44.93 | 39.20 | 44.24 | 2.37 | - | 49.28 |
| SHOT+SO | 20.46 | 9.32 | 12.40 | 9.12 | 12.05 | 14.84 | 34.67 | 11.63 | 9.10 | 35.58 | 10.81 | 7.54 | - | 60.20 |
| TENT+SO | 42.59 | 55.91 | 55.91 | 55.91 | 55.91 | 6.35 | 39.90 | 45.09 | 46.01 | 39.20 | 45.75 | 2.37 | - | 57.15 |
| Avg. | 31.52 | 32.62 | 34.16 | 32.52 | 33.98 | 10.60 | 37.29 | 28.36 | 27.56 | 37.39 | 28.28 | 4.95 | - | 58.68 |
| Avg. Rank | 5.50 | 5.75 | 3.75 | 6.25 | 4.25 | 7.50 | 5.50 | 7.00 | 8.00 | 5.50 | 7.00 | 12.00 | - | - |
| Correlation | 0.28 | 0.22 | 0.25 | 0.22 | 0.29 | -0.18 | 0.11 | -0.05 | -0.02 | -0.03 | 0.02 | -0.53 | - | - |
| Source-only | - | - | - | - | - | - | - | - | - | - | - | - | 56.97 | - |

performance on academic benchmarks. We have also highlighted some surprising results: (i) A strong source checkpoint can be competitive with UDA algorithms when using the RankMe or MMD validators. (ii) Even when validating among both source-only and algorithm checkpoints, performance may be worse than abandoning adaptation altogether and simply using a source-only model as selected by source accuracy – especially in SFDA and TTA. This is a major risk and failure case that will preclude deployment of adaptation in real applications, and one which we encourage future academic work to study. (iii) While TTA algorithms have attracted attention for their strong performance on simple synthetic benchmarks, they fail on more complex distribution shifts such as Office-Home. Future work should carefully consider model selection pipelines, choice of validators, data splits and, at times, whether to perform adaptation at all.

## 6 Limitations

Since this paper covers multiple settings and many algorithms, we were only able to run each algorithm on each domain shift with 10 hyperparameter choices. Ideally, this number would be higher, but adapting many models is expensive, and in reality a practitioner would also face computational limitations. If we had been able to increase this number, the effects would likely be minor improvements in the highest scores achievable by the algorithms, but it would not guarantee that any given validator would choose a better checkpoint. A second limitation is our preliminary study of regression adaptation. Many validators are designed for classification and performance when applied to regression may suffer. However, we view this as a call for study of validation criteria across the full range of adaptation applications from regression to segmentation and detection.

## 7 Broader Impact Statement

As distribution shifts often occur in real-world problems, the use of domain adaptation techniques to tackle them has become commonplace. We have highlighted weaknesses in the standard practice found in DA literature which, if applied in safety-critical scenarios, could lead to catastrophic outcomes. We have therefore outlined a set of better practices for practitioners looking to apply DA to their problems and presented a more realistic view of the field's current potential. Our hope is that this will decrease the likelihood of bad outcomes in its real-world application. The environmental impact of running a large-scale benchmarking study is significant. However, we aim to provide a clear pipeline for future work to use, which can ultimately reduce unnecessary computation due to bad practice.

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

Table 8: Comparison of validation criteria for model selection in UDA for a regression task. We report the target test MSE for the top models selected by each validator. Lower is better. The colour of a cell indicates whether that model/validator combination beats the source-only model (green) or not (red), with a darker red colour meaning it gets more than twice the MSE of the source-only model performance.

| | RankMe | AMI | ARI | V-Measure | FMI | Silhouette | DBI | CHI | BNM | MMD | CORAL | SND | IM | Entropy | MSE | Oracle |
|---|---|---|---|---|---|---|---|---|---|---|---|---|---|---|---|---|
| ADDA | 17.52 | 43.96 | 44.44 | 43.96 | 44.44 | 46.69 | 16.50 | 18.31 | 46.69 | 16.76 | 17.49 | 17.52 | 17.49 | 16.50 | 46.27 | 16.50 |
| CORAL | 45.06 | 45.12 | 46.09 | 45.12 | 46.09 | 45.22 | 44.98 | 46.68 | 47.02 | 47.02 | 47.02 | 45.71 | 47.02 | 47.02 | 47.11 | 43.60 |
| DANN | 81.63 | 42.17 | 42.17 | 42.17 | 46.95 | 79.15 | 54.40 | 55.18 | 74.76 | 81.63 | 53.99 | 35.22 | 81.63 | 54.32 | 45.41 | 35.22 |
| GAN | 16.29 | 42.06 | 43.37 | 42.06 | 43.22 | 53.75 | 47.28 | 49.41 | 47.60 | 16.29 | 16.80 | 53.74 | 40.53 | 31.59 | 44.10 | 16.29 |
| MMD | 64.96 | 46.86 | 46.94 | 46.86 | 46.94 | 90.67 | 55.63 | 138.98 | 138.98 | 64.96 | 63.46 | 45.41 | 138.98 | 90.67 | 61.47 | 45.12 |
| VADA | 17.54 | 44.69 | 57.33 | 44.69 | 57.33 | 48.26 | 17.54 | 41.73 | 51.65 | 14.38 | 14.95 | 46.27 | 14.38 | 55.82 | 48.44 | 14.38 |
| Avg. ↓ | 40.50 | 44.14 | 46.72 | 44.14 | 47.50 | 60.62 | 39.39 | 58.38 | 67.78 | 40.17 | 35.62 | 40.64 | 56.67 | 49.32 | 48.80 | 28.52 |
| Avg. Rank ↓ | 6.33 | 5.42 | 8.42 | 5.42 | 8.58 | 11.33 | 5.50 | 10.00 | 12.58 | 6.92 | 6.25 | 6.25 | 8.50 | 8.33 | 10.17 | - |
| Correlation ↑ | -0.31 | 0.13 | -0.09 | 0.13 | -0.16 | -0.17 | 0.05 | 0.12 | -0.25 | 0.17 | 0.12 | -0.03 | -0.07 | -0.25 | -0.29 | - |
| Source-only | 49.06 | 49.06 | 49.06 | 49.06 | 49.06 | 49.06 | 49.06 | 49.06 | 49.06 | 49.06 | 49.06 | 49.06 | 49.06 | 49.06 | 46.96 | 42.28 |

## A  Regression

**Setup**: We construct a regression dataset with a domain shift akin to MNIST-M [12]. The source domain consists of 32x32 images where, for each image, a single digit taken from MNIST is pasted onto a black background. The digit is randomly scaled between 4x4 and 16x16 pixels and its location is randomised while ensuring the entire digit is visible. The label accompanying the image is the top left $(x_1, y_1)$ and bottom right $(x_2, y_2)$ coordinates of the digit bounding box. The target domain is constructed similarly, but instead of a black background, we use 32x32 regions cropped from the BSDS500 dataset [2].

We discretize the label space as follows. The bounding box labels and predictions take the following form $\{x_1, y_1, x_2, y_2\}$, where each element is a real value between 0 and 1. The function $q$ discretizes each element in the vector into one of 8 uniformly spaced classes between 0 and 1. The class of the full vector is then $c = q(x_1) + 8q(y_1) + 8^2 q(x_2) + 8^3 q(y_2)$. This transformation into class values is performed for all labels and predictions.

The architecture is the same as used for the MNIST-M experiments in the classification setting, with an adjusted final layer for regression. The loss used on the source data is the mean squared error. All other details are the same as in the above UDA setting. For this setup we train six algorithms, ADDA [41], CORAL [39], DANN [10], GAN [13], MMD [42], VADA [38]. Many of the validators we have considered so far rely on categorical labels and predictions. In this regression setup we, therefore, discretize the label space as described above.

**Results**: Table 8 shows the results on this regression task.

**Are conclusions still valid beyond image classification?**     The overall results show a similar trend to the observation of UDA for image classification. 1) Now, CORAL works as the best validator. However, there is no one validator working consistently well for all methods. 2) CORAL, as a UDA method, works most robustly with all validation criteria, leading to all selected results close to its oracle performance, which, though, is not ideal. However, we can see now the correlations are very low for all validators, indicating that there is no reliable validator in this case that works robustly to select a good UDA model.

## B  Assets

**Code**: Our anonymized code base is available at https://anon-github.automl.cc/r/better-da-4936. In this work, we make use of the KevinMusgrave/pytorch-adapt, DequanWang/tent, vita-epfl/ttt-plus-plus and matthijsz/weightedcorr libraries, all available on GitHub and all released under the MIT License.

**Data**: The creators of the MNIST [19], MNIST-M [10] and Office-31 [34] datasets have not provided obvious licenses, but both datasets were created for open academic use. Both VisDA-2017 [29] and

Office-Home [43] are released under custom licenses allowing non-commercial research and use for educational purposes.

## C Training Details

### C.1 Settings

To fully clarify our adaptation settings, we present in algorithms 1, 2 and 3 the benchmarking procedure for UDA, SFDA and TTA, respectively.

---
**Algorithm 1** UDA benchmarking setup.

---
**Require:** Source data $\mathcal{D}_S$, target data $\mathcal{D}_T$, algorithm $\mathcal{A}$, parameters $\boldsymbol{\theta}$, hyperparameter search space $\mathbb{H}$, validator $V$.

    **for** hyperparameters $\boldsymbol{h} \sim \mathbb{H}$ **do**                                ▷ Sample hyperparameters
         $\boldsymbol{\theta}_{\boldsymbol{h}}^* = \arg\min_{\boldsymbol{\theta}} \mathcal{A}(\boldsymbol{\theta}, \mathcal{D}_S, \mathcal{D}_T; \boldsymbol{h})$                        ▷ Optimise model
    **end for**
    $\boldsymbol{h}^* = \arg\max_{\boldsymbol{h}} V(\boldsymbol{\theta}_{\boldsymbol{h}}^*, \mathcal{D}_S, \mathcal{D}_T)$                    ▷ Select best hyperparameters

---

---
**Algorithm 2** SFDA benchmarking setup.

---
**Require:** Target data $\mathcal{D}_T$, algorithm $\mathcal{A}$, parameters $\boldsymbol{\theta}$, hyperparameter search space $\mathbb{H}$, validator $V$.

    **for** hyperparameters $\boldsymbol{h} \sim \mathbb{H}$ **do**                                ▷ Sample hyperparameters
         $\boldsymbol{\theta}_{\boldsymbol{h}}^* = \arg\min_{\boldsymbol{\theta}} \mathcal{A}(\boldsymbol{\theta}, \mathcal{D}_T; \boldsymbol{h})$                            ▷ Optimise model
    **end for**
    $\boldsymbol{h}^* = \arg\max_{\boldsymbol{h}} V(\boldsymbol{\theta}_{\boldsymbol{h}}^*, \mathcal{D}_T)$                        ▷ Select best hyperparameters

---

---
**Algorithm 3** TTA benchmarking setup.

---
**Require:** Test data $\mathcal{D}_T$, algorithm $\mathcal{A}$, parameters $\phi$, hyperparameter search space $\mathbb{H}$, validator $V$.

    **for** each batch $X \sim \mathcal{D}_T$ **do**
        **for** hyperparameters $\boldsymbol{h} \sim \mathbb{H}$ **do**                        ▷ Sample hyperparameters
              $\boldsymbol{\theta} = \phi$                                      ▷ Reset model
              $\boldsymbol{\theta}_{\boldsymbol{h}}^* = \arg\min_{\boldsymbol{\theta}} \mathcal{A}(\boldsymbol{\theta}, X; \boldsymbol{h})$                  ▷ Optimise model
        **end for**
        $\boldsymbol{h}_X^* = \arg\max_{\boldsymbol{h}} V(\boldsymbol{\theta}_{\boldsymbol{h}}^*, X)$                 ▷ Select best hyperparameters
    **end for**

---

**Data splits:** We split all domains of all datasets into train (60%), val (20%) and test (20%) splits.

**Optimisation:** The optimiser for UDA classification, UDA regression, and SFDA is Adam with parameters {betas=(0.9, 0.999)} and weight decay of $1e-4$. For TTA the optimizer is SGD with a momentum of 0.9 (the optimizer is reset after each batch, like the model parameters). The learning rate across all settings is sampled from a log-uniform distribution over [1e-5, 1e-1].

    We train for 100 epochs for MNIST-M and MNIST-MR, 200 on VisDA-2017 and Office-Home. On Office-31 it is 200 if amazon is the target and 2000 otherwise. The number of saved checkpoints is always 20. For our episodic TTA setup, we perform 20 updates on each batch, saving a checkpoint after each update.

**Architecture:** The backbone for experiments on MNIST-M and our regression version MNIST-MR is a LeNet-5[2] [18], and for all other experiments, a ResNet50 [14]. The classifier/regressor is an MLP with two blocks of {Linear, ReLU, Dropout} followed by a final linear layer.

---

[2]The LeNet backbone consists of the convolutional block {Conv, ReLU, MaxPool, Conv, ReLU, MaxPool}.

**Source Training**: The source model consists of the backbone and classifier/regressor as defined above. When using a ResNet50 backbone, we initialise it with ImageNet pre-trained weights (available in PyTorch [27] as `resnet50(weights=ResNet50_Weights.IMAGENET1K_V1)` and freeze the backbone during source training, thereby only updating the classification head. When using a LeNet backbone, we update the entire network during source training. For TTA on CIFAR10-C we use the pre-trained CIFAR10 checkpoint provided by [22] as initialisation. In all other cases, the best model checkpoint as selected by source validation accuracy is used as the initialisation for all adaptation algorithms.

**Adaptation**: During UDA adaptation, the model receives a batch consisting of 64 source examples (with labels) and 64 target examples (without labels). For SFDA, the model only receives the target examples.

## C.2 Hyperparameters

Throughout the experiments conducted in this work, we perform random search for finding the best hyperparameters, with 10 random choices per algorithm. Better performance can potentially be reached by using e.g. BayesOpt. In this work, we focus on analysis and prefer the simpler random search to (1) enable computing a correlation score between the validation criteria and test performance (correlation computed over both high and low-quality checkpoints), as we report in our main tables. Also (2) because our comparisons involve comparing the "best" possible checkpoint with the one discovered by each validator, we did not want to risk aggressively optimising a bad validator and thus having no good checkpoints available for selection by the oracle. The hyperparameter search spaces for all algorithms are specified in Tab. 9. Whenever possible, these are identical to those used in [26].

## D  Validation Details

### D.1  Validators

A recent work systematically investigated the possible validation criteria for UDA, which we summarise below using $\hat{y}$ to denote the one-hot predictions of the model and $y$ as the one-hot ground truth labels. **Source accuracy**: $d$ is simply the accuracy metric and $\mathcal{D}_V$ can be a training or validation set from a source domain.

$$d(f_{\boldsymbol{\theta}}, \mathcal{D}_V) = \frac{1}{N_V} \sum_{i=1}^{N_V} \mathbf{1}(\hat{\boldsymbol{y}} = \boldsymbol{y}), \tag{5}$$

where $\mathbf{1}(\cdot)$ is the indicator function that evaluates to one if its argument is true and zero otherwise. **Entropy**: Entropy has been used in an adaptation loss [45] as well as for model selection. In this case, $d$ computes the confidence of the model predictions, as measured by the entropy of the predicted label distribution, and $\mathcal{D}_V$ is typically the training or validation set from an unlabelled target domain. We further investigate the effect when $\mathcal{D}_V$ comes from the source domain.

$$d(f_{\boldsymbol{\theta}}, \mathcal{D}_V) = \frac{1}{N_V} \sum_{i=1}^{N_V} H(\boldsymbol{p}_i), \boldsymbol{p}_i = f_{\boldsymbol{\theta}}(\boldsymbol{x}_i), \tag{6}$$

where

$$H(\boldsymbol{p}) = -\sum_{j=1}^{K} p_{[j]} \log p_{[j]}, \tag{7}$$

computes the entropy of the categorical distribution, $\boldsymbol{p}$. **Information maximisation (IM)**: IM is often used as an adaptation loss as well [37] to maximise the diversity of prediction in addition to

Table 9: Hyperparameter search spaces for all algorithms considered. Some algorithms are used in multiple settings (e.g. DANN and MMD are used for UDA classification and regression and SHOT is used for SFDA and TTA). In such cases, the search spaces are the same across settings.

| Algorithm | Hyperparameter | Search Space |
|---|---|---|
| ATDOC | $\lambda_{atdoc}$ | [0, 1] |
| | $K_{atdoc}$ | int([5, 25], step=5) |
| | $\lambda_L$ | [0, 1] |
| BNM | $\lambda_{bnm}$ | [0, 1] |
| | $\lambda_L$ | [0, 1] |
| DANN | $\lambda_D$ | [0, 1] |
| | $\lambda_{grl}$ | log([0.1, 10]) |
| | $\lambda_L$ | [0, 1] |
| MCC | $\lambda_{mcc}$ | [0, 1] |
| | $T_{mcc}$ | [0.2, 5]) |
| | $\lambda_L$ | [0, 1] |
| MCD | $N_{mcd}$ | int([1, 10]) |
| | $\lambda_{disc}$ | [0, 1] |
| | $\lambda_L$ | [0, 1] |
| MMD | $\lambda_F$ | [0, 1] |
| | $\gamma_{exp}$ | int([1, 8]) |
| | $\lambda_L$ | [0, 1] |
| ADDA | $\lambda_D$ | [0, 1] |
| | $\lambda_G$ | [0, 1] |
| CORAL | $\lambda_F$ | [0, 1] |
| | $\lambda_L$ | [0, 1] |
| GAN | $\lambda_D$ | [0, 1] |
| | $\lambda_G$ | [0, 1] |
| | $\lambda_L$ | [0, 1] |
| VADA | $\lambda_D$ | [0, 1] |
| | $\lambda_G$ | [0, 1] |
| | $\lambda_V$ | [0, 1] |
| | $\lambda_E$ | [0, 1] |
| | $\lambda_L$ | [0, 1] |
| AAD | $\lambda_{aad}$ | [0, 1] |
| | $K_{aad}$ | int([3, 5] |
| NRC | $\lambda_L$ | [0, 1] |
| | $K_{nrc}$ | int([2, 5] |
| | $KK_{nrc}$ | int([2, 5] |
| SHOT | $\lambda_{cls}$ | [0, 1] |
| | $\lambda_{ent}$ | [0, 1] |
| | $\lambda_L$ | [0, 1] |
| TENT | $\lambda_L$ | [0, 1] |

confidence.

$$d(f_{\boldsymbol{\theta}}, \mathcal{D}_V) = H\left(\frac{1}{N_V} \sum_{i=1}^{N_V} \boldsymbol{p}_i\right) - \frac{1}{N_v} \sum_{i=1}^{N_v} H(\boldsymbol{p}_i). \tag{8}$$

**Adjusted Mutual Information (AMI)**: This is the adjusted mutual information between predicted and cluster labels.

$$d(f_{\boldsymbol{\theta}}, \mathcal{D}_V) = \mathrm{AMI}(\boldsymbol{p}, \mathrm{CL}(\mathcal{D}_V)) \tag{9}$$

where $\mathrm{CL}(\mathcal{D}_V)$ is the cluster labels for validation set $\mathcal{D}_V$, which can be the target training or validation set.

**V-Measure**: Similarly to AMI, this is a metric defined over clustering labels and predictions. It is defined as the harmonic mean between homogeneity and completeness [33].

**Other clustering measures**: Along with AMI and V-Measure, we compute several other related clustering measures, namely, adjusted Rand index, Fowlkes–Mallows index, silhouette score, Davies–Bouldin index and Calinski-Harabasz index.

**RankMe**: Originally proposed for estimating the transferability of self-supervised representations [11], RankMe approximates the rank of the feature matrix on pre-training data. We investigate its application to both source and target domain data.

**CORAL**: CORAL is an adaptation algorithm that aligns the feature distributions of the source and target data by minimising second-order statistics [39]. Their loss can be used as a validator and is defined as the difference between the covariance matrices of the two domains, $C_S$ and $C_T$.

$$d(f_{\boldsymbol{\theta}}, \mathcal{D}_V) = \mathrm{CORAL}(\mathcal{D}_S, \mathcal{D}_T) = \frac{1}{4d^2} \|C_S - C_T\|_F^2 \tag{10}$$

**Maximum mean discrepancy (MMD)**: A common metric used to compute the discrepancy of feature distributions from source and target domains [42], which can be used with the assumption that the trained model may have a good target performance when the source and target domain features are aligned.

$$
\begin{aligned}
d(f_{\boldsymbol{\theta}}, \mathcal{D}_V) &= \mathrm{MMD}(\mathcal{D}_S, \mathcal{D}_T) \\
&= \frac{1}{N_S(N_S - 1)} \sum_{i=1}^{N_S} \sum_{j \neq i}^{N_S} k(\boldsymbol{s}_i, \boldsymbol{s}_j; f_{\boldsymbol{\theta}}) \\
&+ \frac{1}{N_T(N_T - 1)} \sum_{i=1}^{N_T} \sum_{j \neq i}^{N_T} k(\boldsymbol{t}_i, \boldsymbol{t}_j) \\
&- \frac{2}{N_S N_T} \sum_{i=1}^{N_S} \sum_{j=1}^{N_T} k(\boldsymbol{s}_i, \boldsymbol{t}_j), \\
k(a, b) &= \exp\left\{\frac{-\|a - b\|_2^2}{e}\right\},
\end{aligned}
\tag{11}
$$

where $\boldsymbol{s}$ and $\boldsymbol{t}$ are the features extracted for the data from source and target domains, respectively. When MMD is used for validation, the validation set combines the train sets or validation sets of source and target domains. **Soft neighbourhood density (SND)**: SND computes the entropy based on the gram matrix of the validation features.

$$
\begin{aligned}
d(f_{\boldsymbol{\theta}}, \mathcal{D}_V) &= H(\alpha(\boldsymbol{X}, \tau)), \\
\boldsymbol{X} &= \boldsymbol{v}^T \boldsymbol{v},
\end{aligned}
\tag{12}
$$

where $\boldsymbol{v}$ are the data features, $\alpha(\cdot)$ and $\tau$ are softmax function and temperature. Here $\mathcal{D}_V$ can be the train or validation set of source or target domains. **Batch nuclear-norm maximisation (BNM)**:

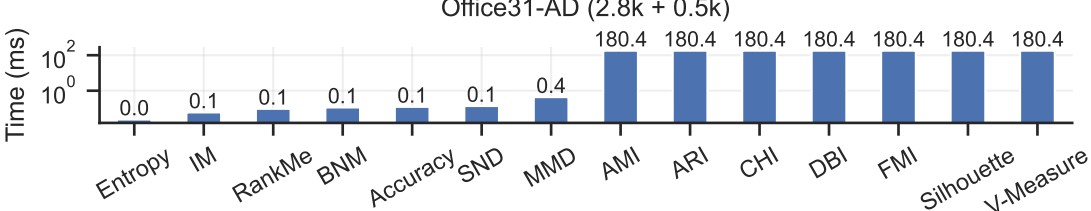

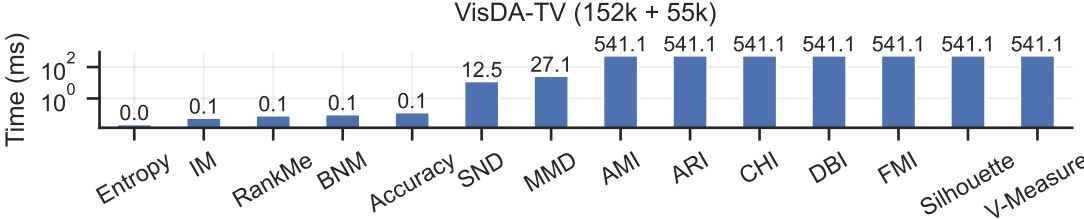

Figure 3: Computation time of UDA validators on (top) Office31-AD and (bottom) VisDA-TV. The clustering-based validators are all significantly more compute-intensive, though it still takes only half a second to compute for a total of 200k datapoints on the VisDA dataset.

BNM was originally a UDA algorithm, which maximises the nuclear norm of the prediction matrix in a batch, being repurposed as a validation criterion.

$$d(f_{\boldsymbol{\theta}}, \mathcal{D}_V) = \|\boldsymbol{P}\|_*, \boldsymbol{P} = f_{\boldsymbol{\theta}}(\mathcal{D}_V) \tag{13}$$

where $\boldsymbol{P} \in R^{N_V \times C}$ the prediction matrix of whole data in $\mathcal{D}_V$ using $f_{\boldsymbol{\theta}}$. And $\|\|_*$ computes the nuclear norm.

### D.2 Time Complexity

Most validators are very quick to compute, requiring only a loop through the features, logits or predictions or some matrix multiplications on the same. Those requiring the extra clustering step (AMI, ARI, CHI, DBI, FMI, V-Measure and Silhouette) all take significantly longer. Nonetheless, no validator is prohibitively expensive compared to the time required to adapt the models. See Fig. 3 for numbers on two representative datasets.

### D.3 Validator Versions in Main Paper

For the tables and figures in the main document, we present a single version of each validator, the one that gives the highest performance when averaged over algorithms and datasets. However, there are multiple options for each, for example, which data split is used or whether we use features, logits or prediction vectors to compute the score. Table 10 shows which versions are used for each validator.

## E  Correlations

Previous works have focused on identifying the validators that have the strongest correlation with the oracle [25, 1]. Our main focus is on finding the ones that select the top-performing models and as we see in the main document, these different methods do not always lead to the same selection. For completeness, we include in Figs. 4 to 7 the weighted Spearman correlations (as used in [25]) of

Table 10: Summary of which version of each validator is presented in the tables of the main document. $S_V$: source val split, $T_T$: target train split, $T_V$: target val split.

| Setting | Option | RankMe | AMI | ARI | V-Measure | FMI | Silhouette | DBI | CHI | BNM | MMD | CORAL | -SND | IM | Entropy | Accuracy/MSE |
|---|---|---|---|---|---|---|---|---|---|---|---|---|---|---|---|---|
| UDA (Classification) | Split | $S_V + T_V$ | $S_V + T_V$ | $S_V + T_V$ | $S_V + T_V$ | $S_V + T_V$ | $S_V + T_T$ | $S_V + T_V$ | $S_V + T_V$ | $S_V + T_T$ | $S_V + T_V$ | $S_V + T_V$ | $T_T$ | $S_V + T_T$ | $S_V + T_T$ | $S_V$ |
|  | Layer | Predictions | Logits | Logits | Logits | Logits | Logits | Logits | Logits | Predictions | Predictions | Predictions | Predictions | Predictions | Predictions | Predictions |
| UDA (Regression) | Split | $T_V$ | $S_V + T_T$ | $S_V + T_T$ | $S_V + T_T$ | $S_V + T_V$ | $S_V + T_T$ | $S_V + T_V$ | $S_V + T_T$ | $T_T$ | $S_V + T_V$ | $S_V + T_T$ | $T_V$ | $T_T$ | $T_T$ | $S_V$ |
|  | Layer | Predictions | Features | Features | Features | Features | Features | Features | Logits | Predictions | Predictions | Features | Predictions | Predictions | Predictions | Predictions |
| SFDA | Split | $T_T$ | $T_T$ | $T_V$ | $T_T$ | $T_T$ | $T_T$ | $T_T$ | $T_T$ | $T_T$ | - | - | $T_V$ | $T_T$ | $T_T$ | - |
|  | Layer | Predictions | Features | Logits | Features | Features | Logits | Logits | Features | Predictions | - | - | Predictions | Predictions | Predictions | - |
| TTA (CIFAR10-C) | Split | $T_T$ | $T_T$ | $T_T$ | $T_T$ | $T_T$ | $T_T$ | $T_T$ | $T_T$ | $T_T$ | - | - | $T_T$ | $T_T$ | $T_T$ | - |
|  | Layer | Predictions | Logits | Logits | Logits | Logits | Logits | Features | Logits | Predictions | - | - | Predictions | Predictions | Predictions | - |
| TTA (Office-Home) | Split | $T_T$ | $T_T$ | $T_T$ | $T_T$ | $T_T$ | $T_T$ | $T_T$ | $T_T$ | $T_T$ | - | - | $T_T$ | $T_T$ | $T_T$ | - |
|  | Layer | Logits | Features | Features | Features | Features | Features | Features | Features | Predictions | - | - | Predictions | Predictions | Predictions | - |

all validators considered. Additionally, the comparison of train and val splits for validators in terms of correlation is shown in Fig. 8.

## F  Compute Resources

The majority of experiments were run on an 8xA6000 internal cluster machine. The total number of algorithms we have trained and validated in this work is  2,440. Assuming on average the training time is 1h per algorithm, this means 2,440 GPU hours have been used.

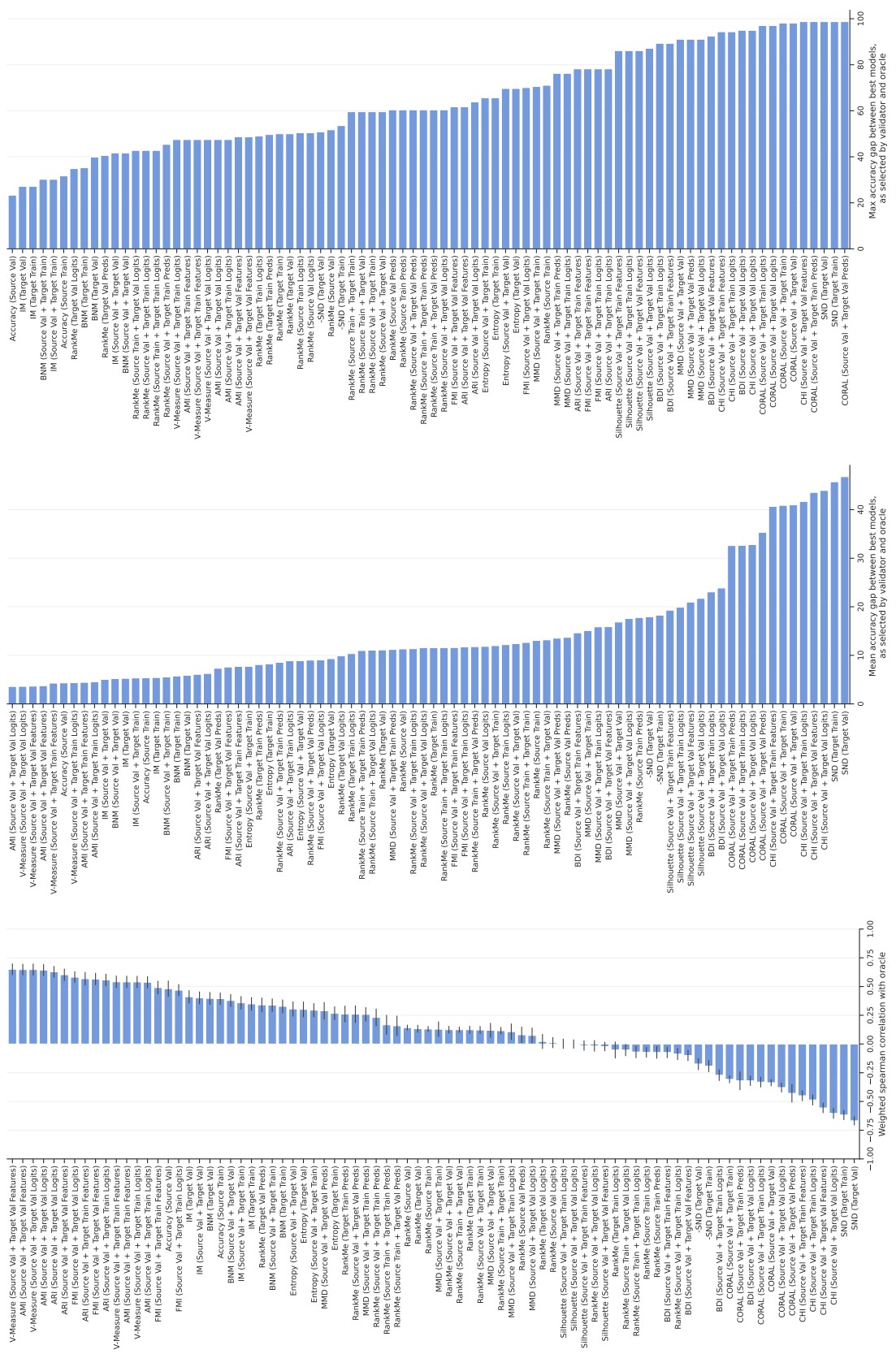

Figure 4: Left: Correlations with target test accuracy on UDA benchmarks. Error bars are standard error across domains. Middle: Average gap between the best model as selected by each validator and the oracle. Right: Maximum gap between the best model as selected by each validator and the oracle.

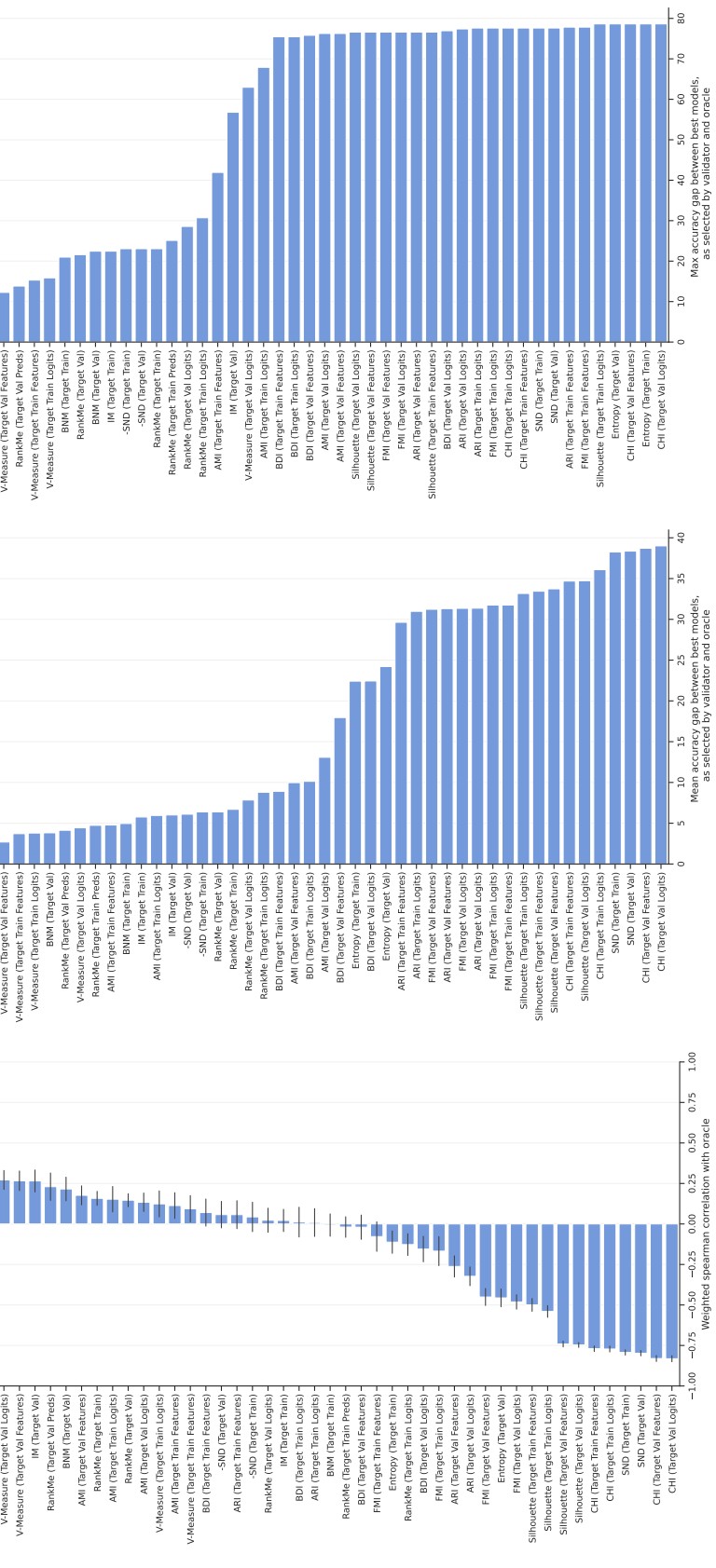

Figure 5: Left: Correlations with target test accuracy on SFDA benchmarks. Error bars are standard error across domains. Middle: Average gap between the best model as selected by each validator and the oracle. Right: Maximum gap between the best model as selected by each validator and the oracle.

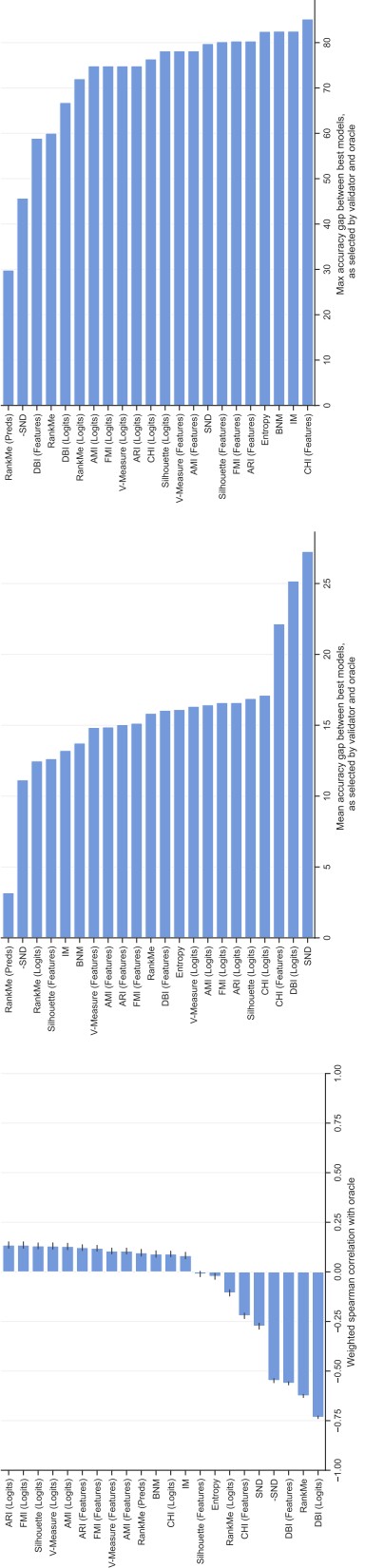

Figure 6: Left: Correlations with target test accuracy on TTA CIFAR10-C. Error bars are standard error across domains. Middle: Average gap between the best model as selected by each validator and the oracle. Right: Maximum gap between the best model as selected by each validator and the oracle.

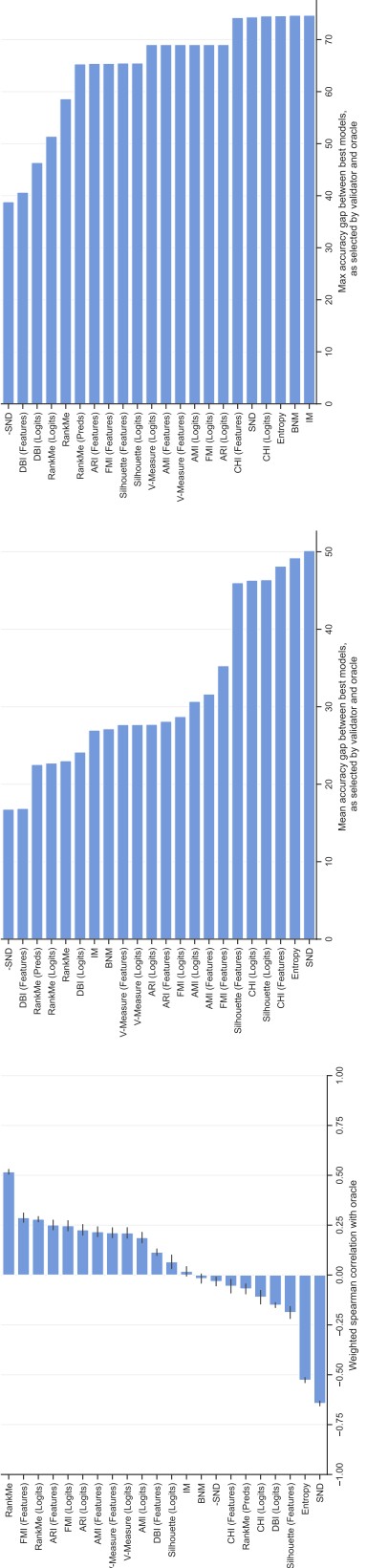

Figure 7: Left: Correlations with target test accuracy on TTA Office-Home. Error bars are standard error across domains. Middle: Average gap between the best model as selected by each validator and the oracle. Right: Maximum gap between the best model as selected by each validator and the oracle.

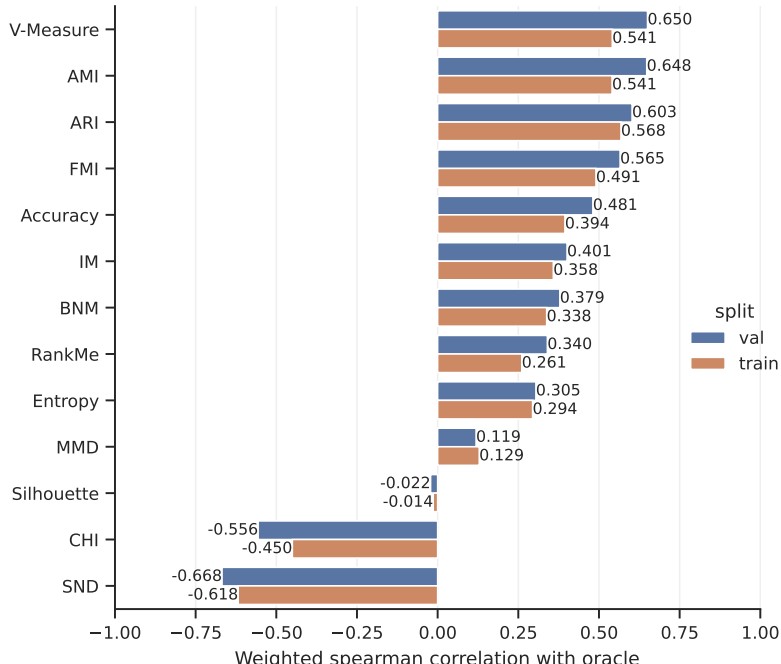

Figure 8: Comparison of split for evaluation of validation criteria in the UDA (classification) setting. We report the average weighted Spearman rank correlation between each validator and target test accuracy when using the following data splits for computing validators: (orange) target train data and (blue) target validation data.

