# OpenReview forum: "Better Practices for Domain Adaptation"
_automl.cc/AutoML/2023/Conference — AutoML 2023 MainTrack_

### Official Review · Reviewer_2u7o · 2023-04-10

**Potential Impact On The Field Of Automl Rating:** 3
**Technical Quality And Correctness Rating:** 4
**Clarity Rating:** 4

**Summary Of Contributions:**

The paper examines the current state of domain adaptation using approapriate evaluation methods by testing various validation criteria and applying them to assess popular adaptation algorithms. It shows that all three branches of domain adaptation consistently face significant challenges. Although attainable performance are generally lower than expected, using appropriate validation splits and metrics is beneficial.


**Actions Required To Increase Overall Recommendation:**

Given the empirical aspect of the paper, it is hard to substantially improve the quality of the paper.

**Clarity:**

The paper is overall well written.  Minor typos could be fixed in the final version of the paper.

**Overall Review:**

Pros:

The authors clearly identify bad practices for HPO in conjunction with DA and motivate the paper in an appropriate way accordingly.
The experimental setup covers a large spectrum for DA scenarios and considers a large variety of validators.
The outcomes of different setups show some general trend despite some inconsistencies which makes them valuable take-home messages.

Cons:

The paper barely fits into the scope of an AutoML conference since hyperparameters are fixed in their respective setups. The promise that the introduced data split combined with an appropriate validator could help reseach on HPO under DA conditions was not sufficiently supported (although it sounds plausible)



**Potential Impact On The Field Of Automl:**

I expected more attention to the V-measure as a validator in the near future since it performs best in many scenarios. Although the V-measure is not new per se, it has not been used as a validator in the domain adaptation literature. The paper could also motivate further investigation on how to make better use of the results toward a robuster HPO in DA context

**Review Confidence:**

4: You are confident in your assessment, but not absolutely certain. It is unlikely, but not impossible, that you did not understand some parts of the submission or that you are unfamiliar with some pieces of related work.

**Review Rating:**

6: Borderline Leaning Accept: Technically sound paper where reasons to accept outweigh reasons to reject. Please use sparingly.

**Review Summary:**

In general, this is a good paper.  The fact that it is a bit off topic and has little to do with AutoML makes it potentially not suitable for this venue. I am willing to upgrade my voting if the author manage to show a stronger connection to the conference topic.

**Technical Quality And Correctness:**

The paper builds strongly on empirical results and I have not identified any flaw with regard to the contributions or claims.

---

> ### Author Response · Authors · 2023-05-01
> **Relevance to AutoML**
>
> Thank you for your comments.
> Please see the [general comment](https://openreview.net/forum?id=tQz8u2KU3zy&noteId=7ZMTnv1Y38) and the updated introduction in the revised paper, both clarifying how our study relates to the AutoML community.

---

### Official Review · Reviewer_Xd7a · 2023-04-12

**Potential Impact On The Field Of Automl Rating:** 3
**Technical Quality And Correctness Rating:** 4
**Clarity Rating:** 3

**Summary Of Contributions:**

This paper identifies the problem that many unsupervised domain adaptation (UDA) works use unrealistic validation methods especially for AutoML, so they propose methods for and perform benchmarks using only validation data that is available in each UDA scenario of increasing constraints.

**Actions Required To Increase Overall Recommendation:**

* Clarify and further analyze your tables of results
* Explicitly propose how to practically apply the findings of this paper

**Clarity:**

The paper is written and organized clearly overall. The language is generally easy to understand.

Analyses of results are often written in numbered lists within a paragraph, with 1-2 sentences each. While this saves a lot of space, it doesn't allow for much development of ideas. Try to expand these points to 2-4 sentences each to ensure each one is well-supported and well-motivated. What are the implications and/or impact of each conclusion?

**Overall Review:**

**Positive aspects:**

* The problem is identified and motivated very clearly.

* A good breadth of datasets (including both simulated and natural domain shifts), data availability constraints, and backbones are studied.

**Negative aspects:**

* While some HPO is performed within each experiment, there is no analysis of how the training recipe (such as the number of source-only checkpoints, number of target hyperparameter sets, or the optimizer and it's fixed hyperparameters) affects the variance of results.

* Could these studies be used to automatically select the best validators (and subsequently the best algorithms) for a given UDA scenario? This could just be proposed within this paper to help improve the impact.

* The title is a bit too broad. Perhaps something like "Better Practices for Validation of Unsupervised Domain Adaptation" would be better.

* Can you further analyze the "+SO" results? Why did you decide to include the source-only model, and when does it affect the results? How realistic is this setting? Like Table 6, can you add results for the algorithms without the source-only model?

* Why are certain rows/columns mostly blank in Tables 4-6?

* What happens outside of image classification? Are there other relevant tasks and data modalities that could be benchmarked to see if your claims extend beyond image classification? In Section 6, you mention a "preliminary study of regression adaption" but don't seem to actually include these results.

**Minor comments:**

* Within-paragraph numbering is inconsistent: sometimes numbers (1),(2),(3) are used, other times numerals (i), (ii), (iii) are used.

* In line 24, you state "a different conclusion", but some of your conclusions are similar.

* Explicitly mention how to find your code in the main paper, in case the reader does not see it in the checklist.


**Potential Impact On The Field Of Automl:**

This paper's impact and chances of citation are already good but could be further increased by proposing how to practically use the results, such as how to incorporate such approaches into an AutoML pipeline to select the best UDA algorithm(s) with the best validators for a given scenario. You imply in lines 322-323 that this is your intention, so explicitly describe this for your readers.

**Review Confidence:**

3: You are fairly confident in your assessment. It is possible that you did not understand some parts of the submission or that you are unfamiliar with some pieces of related work.

**Review Rating:**

7: Weak Accept: Technically sound paper with moderate-to-high impact and strong evaluation, with perhaps some minor flaws.

**Review Summary:**

This paper is a clear and impactful study on valid validation of UDA techniques. It could be further supported by additional analysis detailed above. I think it is worth accepting, especially if the recommended actions are taken.

**Technical Quality And Correctness:**

All technical aspects of this work seem correct, without any major problems or gaps. This paper seems to cite the necessary relevant works, although I am not an expert in the UDA sub-domain of domain adaptation.

---

> ### Author Response · Authors · 2023-05-01
> **Training recipe and clarifications on tables**
>
> Thank you for your comments.
>
> ### On training recipe:
> With more checkpoints etc, we would expect that the oracle performance level and the best validators would improve, but the worse ones would not. So stretching of the results range. But we don’t expect any fundamental conclusion change.
> We have prioritised a large number of datasets over increasing the number of hyperparameter sets etc.
>
> Thank you for your comments.
>
> ### On automatically selecting the best validator:
> No, this would not be possible. We can’t automatically pick a validator, because this requires test performance. But our empirical results can be used to indicate a good validator overall (since we average over several datasets, the validator shown to be good is likely to generalise to new ones).
>
> ### On the title:
> This is a good suggestion, and we will consider updating the title.
>
> ### On the +SO results:
> If you exclude the source-only checkpoint, sometimes performance is much lower than the baseline. We decided to include it to test the hypothesis that the validator could pick the source-only model as a backup option. We included it for Tab 6 specifically, because this is the one where the performance is worse than baseline. It didn’t make much difference in the other tables.
>
> ### On blank rows/columns in tables:
> Row: We only have a single source-only model available in the SFDA/TTA settings, so there it doesn’t make sense to apply validators to a pool. So you can consider the printed value to apply to the whole row.
> Column: Source validation accuracy requires access to source data. We don’t have this for TTA/SFDA.
>
> ### On other modalities:
> The regression results we discuss are included in the appendix Sec. A. We could benchmark other modalities, but this is out of scope of our current study.

---

> > ### Comment · Reviewer_Xd7a · 2023-05-05
> > **Response to the authors**
> >
> > I appreciate your responses to my concerns. However, few changes were affected to the article itself, so I will maintain my rating.

---

### Official Review · Reviewer_1Bfh · 2023-04-12

**Potential Impact On The Field Of Automl Rating:** 4
**Technical Quality And Correctness Rating:** 4
**Clarity Rating:** 3

**Summary Of Contributions:**

In this paper, the authors analyze the general field of Domain Adaptation by benchmarking a suite of candidate validation criteria and then use these to compare popular adaptation algorithms for the purpose of hyper-parameter optimization and model selection. They consider different branches of domain adaption, Unsupervised Domain Adaptation (UDA), Source free domain adaption (SFDA), and Test time augmentation (TTA) because there seems to be a lack of consistency amongst the validation criteria used to compare different models. In the analysis of methods, covering 15 validators each with 10 hyper-parameter choices, they find that previously unexplored validators (they bring up RankMe and V-Measure) consistently perform well but generally, the best validator is setting/algorithm dependent.

**Actions Required To Increase Overall Recommendation:**

I am fairly confident that Accept is the right evaluation for this piece of work. I have a few questions that I listed in the cons but overall think that they would not cause me to raise my score unless the authors were able to perform an even larger hyper-parameter suite to make their results even stronger and maybe tighten up the writing to make the paper less dense to read.

**Clarity:**

The paper is rather challenging to get through as it is very dense, mostly containing text with only a few figures throughout. Maybe the required thoroughness of such an analysis requires a lot of explanation, but the messages might be better received if some things were cut out and put into tables or additional figures in the main text. Beyond this there are a few minor comments:

- I found that all figures from pages 20-24 being rotated made them incredibly difficult to read. I understand that there is a lot of information to convey but for longer text horizontal is always preferred.
- I might've missed this, but was there any description in the paper regarding what exactly the splits percentages were?

**Overall Review:**

Pros
+ The author choose a very broad and thorough sweep of methods within UDA, SFDA, and TTA as well as a large sweep of validators.
+ The questions both asked and the results answered in the analysis section do a good job of answering the high-level question in the paper regarding the evaluation of the UDA methods.
+ The final investigation of TTA seems very strong in particular. From the analysis surrounding the statement "we suggest that the strong success of TTA methods on synthetic benchmarks may not be representative of real-world adaptation problems", this would be a major contribution that changes how future TTA methods are developed.

Cons
- How to read the correlations is confusing. It seems to me like the correlation of even the final methods (RankMe and V-Measure) the authors propose doesn't seem to be that high. I was wondering where the authors drew the conclusions that these were good validators from.
- Not a con necessarily but something that I felt might be incomplete was the analysis of tuning sensitivity. Tuning sensitivity was mentioned in the results section 4.1.2 but then was not addressed in 4.2.1 or 4.3.1. How does tuning sensitivity compare to the TTA or SFDA regimes?

**Potential Impact On The Field Of Automl:**

As the authors do point out, domain adaptation, and in particular unsupervised domain adaptation, is a crucial paradigm within AutoML because addressing how to handle the issue is what allows for machine learning systems to be deployed in real-world scenarios. Additionally, because there is such inconsistency in validation between different methods, there exists a gap (that the author address) between what methods claim and their actual performance. I think this work stands to be very impactful because follow-up work in the field might be inspired to consider more fair evaluation paradigms and better validators for Hyper-parameter optimization.

**Review Confidence:**

4: You are confident in your assessment, but not absolutely certain. It is unlikely, but not impossible, that you did not understand some parts of the submission or that you are unfamiliar with some pieces of related work.

**Review Rating:**

8: Accept: Technically sound paper with major impact and strong evaluation, with perhaps some minor flaws.

**Review Summary:**

Based on my points above, I think that this is a very solid piece of work that merits acceptance to the conference. Their effort to make a common ground for evaluation and thorough investigation of different methods and validators is definitely accepted paper quality work. I believe as the authors point out that they could potentially sample more hyper-parameter settings for each algorithm because, with the minor improvements incurred, might choose better checkpoints.

**Technical Quality And Correctness:**

From my perspective, the authors have done a great job in being thorough about both their formulation of the problem of model selection and evaluation. I don't believe there are any major problems in the authors technical formulation.

---

> ### Author Response · Authors · 2023-05-01
> **Correlations, tuning sensitivity and more**
>
> Thank you for your feedback.
>
> ### On correlations:
> (1) Yes, it is true that most validators have low correlations. This is to some extent an open problem, which is one of our take home messages. We also consider correlation not to be a great measure of use as a metric to drive HPO, since most validators seem to exhibit non-linear relationships with test performance. We instead think it better to look at final selection performance, which is what we focus most discussion on. We report which the best validators are for each setting, but agree that often they are not strong enough to recommend for practical use (in e.g. TTA).
>
> ### On tuning sensitivity:
> In order to get a better picture of tuning sensitivity, we now include tables reporting the "percentage of results that are equal or better than no-adaptation source-only" for both algorithms (independent of validator) and validators (independent of algorithms). These can be found in Tables 7 & 8 in the revised paper.
>
> ### On data split sizes:
> We split all domains of all datasets into train (60%), val (20%) and test (20%) splits. This has been added to Sec C.1.
>
> ### On rotating figures:
> We will edit these figures to make sure they are large enough to read without rotating.

---

> > ### Comment · Reviewer_1Bfh · 2023-05-01
> > **Thank you for your response.**
> >
> > Thank you for your response. This satisfies the questions I had and I think these additions made the already good paper stronger.

---

### Official Review · Reviewer_UUAd · 2023-04-13

**Potential Impact On The Field Of Automl Rating:** 3
**Technical Quality And Correctness Rating:** 4
**Clarity Rating:** 3
**Actions Required To Increase Overall Recommendation:** 1. Most importantly I'd like to under…

**Summary Of Contributions:**

The paper introduces a benchmark for domain adaptation techniques across unsupervised domain adaptation, source-free domain adaptation, and test-time adaptation that focuses on ensuring that the evaluation methods are aligned with practical application by avoiding using test data labels for tuning / model selection when in practice these labels would not be available. Within this context, the paper evaluates a variety of validators and adaptation algorithms across various datasets and source-model hyperparameters to identify the combinations that perform well in practice, showing that while strong results can potentially be obtained via UDA, that results for SFDA and TTA are mixed, with high volatility and many algorithms underperforming the no-adaptation baseline. The paper additionally highlights that using a separate target validation set for selection / tuning rather than purely the target train set results in better performance in the UDA setting.

**Clarity:**

The paper is generally well written and easy to follow, with informative tables and a detailed appendix.

One part of the paper I would like clarification on is the emphasis on using a separate train/val split on the target data to achieve superior performance as shown in Figure 2, however this also has its own share of questions left unanswered: How does one choose the split proportions? Wouldn't you need to eventually predict on all of the target data anyways? In that sense, how can you have a validation set without doing cross-validation? Is having separate "train", "val", and "test" sets for the target data realistic (ref Figure 1)? The authors make a major point of this being a key contribution (L227-228), yet I am unsure if it is realistic to have such data available without cross-validation.

minor comment, does not impact my score - L244: "Here, the best validators are RankMe and V-measure" -> RankMe has a worse average rank and correlation than AMI, so it appears you are considering "Avg." to determine what is "best". This observation is inconsistent with the use of "Avg. Rank" to claim the best validator in UDA (AMI > V-Measure in "Avg." in UDA)

**Overall Review:**

The paper provides a useful empirical analysis of DA techniques and reports important negative results that should be taken into careful consideration when thinking about integrating DA into AutoML systems. The paper additionally highlights that using a separate target validation set for selection / tuning rather than purely the target train set results in better performance in the UDA setting.

A lack of multiple random seeds does hinder the analysis, as well as there being little analysis on any "auto" component to assist AutoML in applying DA practically. In particular, I am still confused on the train/val target set details, which I hope is clarified during the rebuttal. Overall, I am borderline leaning accept, and will consider updating my score once my key questions to the authors are addressed.

**Potential Impact On The Field Of Automl:**

Domain adaptation methods could be applied as a component of AutoML systems to enhance accuracy on shifted data. Being able to identify an empirically consistent and robust adaptation strategy would be of significant interest to the AutoML community, and enable AutoML to expand its use to these challenging scenarios.

The paper as it stands isn't directly related to AutoML. While hyperparameter tuning is done, there isn't analysis on how the tuning impacted the final results (for example, the result's sensitivity to the amount of tuning done). While the reported tables can serve as recommendations for the pairing of algorithm and validator, I struggle to believe this alone is enough evidence for AutoML systems to adopt any one particular pairing as a golden standard, and the paper does not make recommendations for how one could automate the selection of such an ideal pairing (which is perhaps a core point of the paper and the present reality of the DA field). In this sense I am ok with that take-away, as a negative result is very useful when considering these approaches from an AutoML standpoint.

**Review Confidence:**

3: You are fairly confident in your assessment. It is possible that you did not understand some parts of the submission or that you are unfamiliar with some pieces of related work.

**Review Rating:**

6: Borderline Leaning Accept: Technically sound paper where reasons to accept outweigh reasons to reject. Please use sparingly.

**Review Summary:**

Overall, I am borderline leaning accept, and will consider updating my score once my key questions to the authors are addressed.

**Technical Quality And Correctness:**

The experiments have no obvious flaws, and the provided code appears reasonable to be able to reproduce the findings. Some limitations exist due to the low number of datasets used (for SFDA and TTA) and the lack of multiple random seeds, thus making any take-aways hard to trust with statistical confidence.

An additional metric that could be useful for readers would be "percentage of results that are equal or better than no-adaptation source-only" for each validator/algorithm, along with statistical significance. This would help identify if any strategy consistently beat no-adaptation, which is perhaps of more interest than strategies that vary wildly in whether they help or harm the test score.

Another metric could be the average variation of the score of a given option, as an indicator of its volatility and magnitude of impact on the score.

---

> ### Author Response · Authors · 2023-05-01
> **Reasoning behind val splits and measures of stability**
>
> Thank you for bringing up a few important points for us to clarify.
>
> ### On train/val/test splits:
> You bring up a good point. There are cases where we would not recommend using a val split.
>
> We can consider two settings, a transductive case where we wish to evaluate our method on all target data given to us or an inductive case where we assume more target data is coming, which makes generalisation more important.
>
> In the transductive case, you may prefer not to use a validation split. We would argue, however, that the inductive case is more realistic, as unlabelled data is often widely accessible in the modern age and organisations often receive continuous streams of data without any clear cut-off points. Due to the importance of generalisation in this setting, using a val split becomes more important.
>
> Additionally, the usefulness of a val split depends on the amount of target data available. We agree that for a bounded-size benchmark, making a train/val split out of the actual train set costs some data, which will give a performance penalty if the benchmark is small. On the other hand, we show that the quality of the validator improves with a val split. So we end up with a trade-off between adaptation and validation quality. At a certain point, the gains from using more data for adaptation diminishes and become more useful for validation.
>
> Overall, we would recommend a val split if we are in the inductive setting and the dataset size is not small.
>
> ### On the stability of algorithms and validators:
> We now include tables reporting the "percentage of results that are equal or better than no-adaptation source-only" for both algorithms (independent of validator) and validators (independent of algorithms). These can be found in Tables 7 & 8 in the revised paper. These will help give an idea of the stability of an algorithm or validator.
>
> ### On relevance to AutoML:
> Please see [general comment](https://openreview.net/forum?id=tQz8u2KU3zy&noteId=7ZMTnv1Y38).

---

> > ### Comment · Reviewer_UUAd · 2023-05-01
> > **Response to Authors**
> >
> > Thank you for your response and updates! Regarding adjustments to my score, the primary remaining point that gives me hesitance is the relevance to AutoML, which I reply with more detail below.
> >
> > ## On train/val/test splits:
> >
> > Thank you for your answer which generally covers my questions. Below I provide some additional thoughts:
> >
> > I would note that from the AutoML perspective, it is likely that users would often not initially try the systems in the inductive setting for specialized scenarios, as the inductive setting is more complex than the transductive setting and the user might not understand how to properly separate train and test data. This confusion can be avoided by implementing cross-validation in order to leverage all data as both train (+val) and test, which removes the need for the user to understand this aspect, and enables the technique to be applied in all scenarios regardless if there is additional data available beyond the test data.
> >
> > Future work investigating the benefits of cross-validation techniques to more efficiently use the available data for both inductive and transductive settings would be very interesting and significantly raise the relevance and applicability to AutoML, but I do not expect this investigation to be part of this work, as it requires re-running all experiments.
> >
> > ## On the stability of algorithms and validators:
> >
> > Thank you for adding these tables! One comment I have is that while these numbers are useful, they are not representative of the true per-task volatility of the algorithms/validators. Instead, they are based on "Averages over all 21 domain
> > transfers evaluated". It would be more useful to have the percentages in the table be based on the success rate on 21*6 tasks, rather than 6 averaged tasks, as in practice, the user will care about the success rate on a single task and not on the average of many tasks.
> >
> > As an example, if an algorithm has a 55% success rate per-task, when averaging over 21 tasks this would likely look like a 100% success rate. The 55% success rate value is more relevant than the 100% success rate value, and should be the value reported.
> >
> > Also, is there a reason success rate is evaluated separately for algorithm and validator? Wouldn't it be reasonable to evaluate success rate on all algorithm+validator pairings? This should then result in a table of identical dimensions to Table 3, but with success rate across 21 tasks for each algorithm+validator pair instead of average accuracy across 21 tasks.
> >
> > ## On relevance to AutoML:
> >
> > While I agree with the authors that the goal they describe is relevant to AutoML, there is still lacking a consistent messaging as to the recommendation to the AutoML community based on the results.
> >
> > My personal take-away is to currently avoid DA in AutoML except for UDA. The choice of optimal general purpose algorithm + validator to use remains unclear, however the paper does a reasonable job of showcasing techniques that should be avoided (Ex: CORAL, SND, etc.). For UDA, the best practice would involve a deeper dive than what was shown in the paper to identify a strategy that minimizes worst-case scenario results while often generating strong results.
> >
> > While this is my personal take-away as someone heavily involved in the AutoML field who would be a direct user of such DA techniques from a system integration perspective, it is an implied take-away and not one that the authors explicitly discuss nor recommend in their conclusion. If the authors proposed some approach-acceptance mechanism to say that a given technique is superior to Source-only-accuracy with some level of statistical certainty, and then recommend a mechanism in the conclusion or analysis sections, this would increase the impact and relevance to AutoML.
> >
> > As a first step, I would recommend the authors filter out obviously bad methods via some criterion to only focus on the performant solutions (for example, filter out methods that are statistically significantly worse than source-only-accuracy), and then conduct more detailed comparisons & statistical tests to come up with a single recommendation to the AutoML community that is actionable and implementable in modern AutoML systems as a best practice.

---

> > > ### Author Response · Authors · 2023-05-02
> > > **Making actionable recommendations for the community**
> > >
> > > We thank the reviewer for their helpful feedback from a perspective within the AutoML community.
> > >
> > > In the new tables, the algorithm success rates are in fact reported in the more fine-grained style you recommend. But the validator success rates are indeed not as helpful as they could be. We will update this.
> > >
> > > We especially appreciate the points about how to provide more direct recommendations for the community, such as performing statistical tests to single out an algorithm/validator pairing for realising AutoML for DA right now. An excellent suggestion we will dig into. This should reveal when such applications are beneficial (likely UDA) and when they are likely to fail (TTA).

---

### Official Review · Reviewer_jnWR · 2023-04-13

**Potential Impact On The Field Of Automl Rating:** 2
**Technical Quality And Correctness Rating:** 3
**Clarity:** This paper is well-written and easy t…
**Clarity Rating:** 4

**Summary Of Contributions:**

This paper studies how to perform HPO for domain adaptation algorithms where the labeled validation set cannot be accessed and provides an empirical study of HPO and model selection for domain adaptation problems.

**Actions Required To Increase Overall Recommendation:**

If the authors can address my concerns in the "Overall Review" part, I am willing to raise my score.

**Overall Review:**

1. This paper provides an empirical study of HPO and model selection for domain adaptation problems and shows some useful findings for domain adaptation. But this paper does not have much potential impact on the AutoML field.
2. It seems the HPO method used in this paper does not been introduced. Is it Eq. (4)? Why not use Bayesian HPO methods?
3. A typo in Line 563.
4. Different validators in Tab. 2 have different time complexity. It is better to consider the time complexity when performing HPO for domain adaptation problems.

**Potential Impact On The Field Of Automl:**

This paper provides an empirical study of HPO and model selection for domain adaptation problems. Thus, it has a large potential impact on the domain adaptation field rather than the AutoML field.

**Review Confidence:**

3: You are fairly confident in your assessment. It is possible that you did not understand some parts of the submission or that you are unfamiliar with some pieces of related work.

**Review Rating:**

6: Borderline Leaning Accept: Technically sound paper where reasons to accept outweigh reasons to reject. Please use sparingly.

**Review Summary:**

I think this paper provides some useful empirical findings for domain adaptation but limited impact on the AutoML field. I lean to accept this paper but I don't object to reject it either.

**Technical Quality And Correctness:**

This paper provides an empirical study and the experimental settings are fair and reasonable.

---

> ### Author Response · Authors · 2023-05-01
> **Clarifications on HPO and time complexity of validators**
>
> Thank you for your feedback.
> ### On HPO:
> We use random search (Detailed in Sec C.2).
> Better performance can potentially be reached by using e.g. BayesOpt like you mention. In this work, we focus on analysis and prefer the simpler random search to (1) enable computing a correlation score between the validation criteria and test performance (correlation computed over both high and low-quality checkpoints), as we report in our main tables. Also because (2) our comparisons involve comparing the “best” possible checkpoint with the one discovered by each validator, we did not want to risk aggressively optimising a bad validator and thus having no good checkpoints available for selection by the oracle.
>
> This clarification has now been added to Sec C.2.
>
> ### On validator runtime:
> We now include figures in the appendix Sec D.2, measuring clock time of validators on two datasets of different sizes. This shows how the clustering-based validators are more compute-intensive but not prohibitively so compared to the cost of adaptation.
>
> ### On relevance to AutoML:
> Please see [general comment](https://openreview.net/forum?id=tQz8u2KU3zy&noteId=7ZMTnv1Y38).

---

### Review · Reproducibility_Reviewer_2vNn · 2023-04-17

**Completeness Of Code And Dataset Supplement Rating:** 4
**Usability And Ease Of Reproducibility Rating:** 4
**Actions Required To Increase The Reproducibility And Overall Recommendation:** N/A

**Completeness Of Code And Dataset Supplement:**

The code and datasets are complete and sufficient.

**Overall Reproducibility Review:**

The code was well organized and the results were well reproduced. I believe this paper's benchmark framework can bring contributions to the community.

**Review Confidence:**

4: You are confident in your assessment, but not absolutely certain. It is unlikely, but not impossible, that you did not understand some parts of the submission or that you are unfamiliar with some pieces of the code or data.

**Review Rating:**

9: Strong Accept, all aspects of this are easily reproducible.

**Review Summary:**

The code is pretty good. I suggest Accept as a reproducibility Reviewer.

**Summary Of Necessary Code And Dataset Supplement:**

The main experiments benchmarked a suite of candidate validation criteria and used them to assess popular adaptation algorithms. Its evaluation extends the benchmark of [1]. The authors of the paper trained a large number of models across several datasets, algorithms and hyperparameter choices to compare different validation criteria. The authors test on three branches of domain adaptation methodology including Unsupervised Domain Adaptation (UDA), Source-Free Domain Adaptation (SFDA), and Test Time Adaptation (TTA).

[1] Kevin Musgrave, Cornell Tech, Serge Belongie, Ser-Nam Lim, and Meta Ai. Unsupervised Domain 376
Adaptation: A Reality Check. In ECCV, 2020

**Usability And Ease Of Reproducibility:**

The code is well-organized and easy to understand. I run some experiments using the provided code following the readme and it indeed get the same results as the paper reports.

---

### Author Response · Authors · 2023-04-26
**Relevance to the AutoML community**

We were surprised the reviewers thought the relevance was unclear. In order to automate anything (algorithm, hyperparameter, checkpoint selection), we need a metric to optimise. In the case of supervised learning, everyone agrees on this metric (validation performance), but it’s not straightforward for UDA/SFDA/TTA due to the lack of labels. The whole point of this paper is to clarify what the AutoML optimisation metric should be for domain adaptation. Once this is clarified, all the rest of AutoML tooling (BayesOpt, etc) can be directly applied. Thus this paper is about laying the foundations that allow bringing AutoML to DA.

Towards this, we have identified some problem settings and metrics where the validators work sufficiently well such that AutoML might work already today (UDA); and some others where the validation metrics are insufficient, meaning AutoML can’t work until better validators are developed (TTA on big domain shift).

We are happy to update our text to better motivate this contribution.